# Dynamic spatial coding in parietal cortex mediates tactile-motor transformation

Janina Klautke[1,7], Celia Foster [2,3,7], W. Pieter Medendorp[4] & Tobias Heed [2,3,5,6]

Movements towards touch on the body require integrating tactile location and body posture information. Tactile processing and movement planning both rely on posterior parietal cortex (PPC) but their interplay is not understood. Here, human participants received tactile stimuli on their crossed and uncrossed feet, dissociating stimulus location relative to anatomy versus external space. Participants pointed to the touch or the equivalent location on the other foot, which dissociates sensory and motor locations. Multi-voxel pattern analysis of concurrently recorded fMRI signals revealed that tactile location was coded anatomically in anterior PPC but spatially in posterior PPC during sensory processing. After movement instructions were specified, PPC exclusively represented the movement goal in space, in regions associated with visuo-motor planning and with regional overlap for sensory, rule-related, and movement coding. Thus, PPC flexibly updates its spatial codes to accommodate rule-based transformation of sensory input to generate movement to environment and own body alike.

We frequently make reaching movements to tactile stimuli, for example to scratch an itch, move a hair away from our face, or to brush off an insect that crawls along our arm. We perform these movements with ease even though the brain must perform complex computations to successfully complete them. Tactile stimulation excites receptors in the skin, and thus tactile location is initially coded relative to the skin's layout as is evident, for instance, in the homuncular organization of primary somatosensory cortex (S1)[1,2]. Yet, the skin is but a 2D sheet wrapped around our 3D body which, in addition, constantly changes its layout when the body parts move. Therefore, converting touch into a spatially-guided motor act presumably requires neural transformations.

The nature of these transformations between different spatial codes has been debated. A popular conceptualization draws tight parallels between visuo-motor and tactile-motor processing. In this view, planning movement towards a touch involves the computation of tactile location from skin-based, anatomical coding into a 3D spatial code –referred to as remapping of touch from a somatotopic or anatomical into an external(-spatial) reference frame– based on the integration of body knowledge and posture information[3-6]. A potential consequence may be that visual and tactile stimuli are represented in the same spatial code, which may afford direct integration of stimuli from the different modalities as well as common further processing regardless of the original sensory modality of the cue[7].

Under this premise, research based on visuo-motor paradigms is an obvious starting point for establishing the principles that underly tactile-motor processing. Studies on visuo-motor transformation have established that PPC encodes visual targets in 3D reference frames where visual objects are coded relative to a specific body part such as the eyes (or gaze direction), head, torso, or a hand, and sometimes in a combination of such codes[8]. Posterior PPC regions reportedly code hand reaches in a gaze-centered reference frame[9-16] while anterior PPC

[1]Biological Psychology and Neuropsychology, University of Hamburg, Hamburg, Germany. [2]Biopsychology & Cognitive Neuroscience, Bielefeld University, Bielefeld, Germany. [3]Center of Excellence in Cognitive Interaction Technology (CITEC), Bielefeld University, Bielefeld, Germany. [4]Radboud University, Donders Institute for Brain, Cognition and Behaviour, Nijmegen, The Netherlands. [5]Cognitive Psychology, Department of Psychology, University of Salzburg, Salzburg, Austria. [6]Centre for Cognitive Neuroscience, University of Salzburg, Salzburg, Austria. [7]These authors contributed equally: Janina Klautke, Celia Foster. ✉e-mail: tobias.heed@plus.ac.at

regions encode them in a hand-centered reference frame[17–24]. In the macaque ventral intraparietal cortex, visual receptive fields are linked to tactile receptive fields, such that the visual space relevant to a given neuron changes when the monkey moves the body part to which the tactile receptive field is linked[25–27]. This coding implies that visual space is flexibly mapped to body locations based on postural information. However, the reference frames employed by parietal regions are often not fixed but can vary depending on the available information or task requirements[10,28–30]. Thus, spatial coding in this cortical region is dynamic, and such dynamics have been demonstrated not only between different contexts but also during the progression of single trials, marking PPC as a key region of sensorimotor transformation.

In the case of touch, the somatotopic organization evident in S1 extends into the anterior regions of PPC in both macaques[31] and humans[32,33]. In contrast, magneto- and electroencephalographic brain signals over parietal cortex are modulated by both the somatotopic and external-spatial location of tactile stimuli[34–36], yet the spatial resolution of these methods limits conclusions about which brain regions use the respective spatial codes. Generally speaking, somatotopic coding appears to occur more anteriorly than external coding. Transcranial magnetic stimulation (TMS) has also provided evidence that the external location of touch affects tactile processing in PPC[37–39]. Yet, even if the relevance of somatotopic and external-spatial coding for touch in PPC appears established, it has remained elusive which parietal regions employ which code and how transformations between them occur.

There are further differences between movement towards tactile targets as compared to movement to visual targets. During reaches towards a visual target, body-related processing is primarily concerned with coordinating the reaching effector for the movement. In contrast, reaching towards a tactile stimulus is more complex in that it requires processing body information about the movement target, as well as relating the tactile target and moving effector, with both belonging to the same body. One puzzle about this double role of the body being the movement target and the moving entity is that touch, proprioception, and movement processing all recruit the posterior parietal cortex (PPC). To date, there is little knowledge about how these sensory and motor roles of PPC are integrated during self-directed actions.

Anterior PPC is involved in coding posture and body configuration in the context of reaching tasks without vision[40–42]. Moreover, a frontoparietal network, including a wide area of PPC, encodes proprioceptive reach targets in a body-centred code[10,29]. It is noteworthy, however, that paradigms investigating movement planning to proprioceptive targets usually name the target body part (e.g., "move to the tip of the left index finger"). Therefore, this type of task does not involve any transformation of tactile information from skin into space or the mapping of touch to a body part. In other words, such tasks test only one aspect of somatosensation, namely proprioception, but do not speak to the processing of touch. Studies that have addressed touch have demonstrated that PPC mediates the sensorimotor transformation of tactile targets on the hands[34,43,44]. However, the respective studies did not dissociate sensory and motor spatial information and so it remains unclear whether the observed responses were related to maintained sensory processing or to movement planning.

We have so far addressed direct spatial transformations, which transform the location of a sensory stimulus into a code that guides the motor system to that location. Such transformations mediate directly between sensory and motor reference frames. However, parietal cortex is also involved in deriving movement plans from arbitrary, abstract sensory cues[45–47]. In such paradigms, the sensory stimulus and movement goal are not identical in their location. For instance, movements depend on rules when a movement must not be directed to a visual cue itself but, instead, a different movement goal must be inferred based on the cue's location. A typical paradigm is the so-called anti-movement task, in which participants must plan a response to a location opposite to that of a (typically visual) cue[48–52]. In this case, the movement goal location is at a different spatial location than the sensory cue, which allows brain responses related to sensory processing and movement planning to be distinguished. In this type of task, the location of the sensory cue is still relevant for solving the task, even if the resulting motor response is not directed towards it. In line with these requirements, experiments that employed this paradigm have established that PPC neurons first encode the location of a visual cue, but later switch to encoding the location of the movement goal that derives from it[49,53]. Human PPC also exhibits such dynamic coding, with responses to visual cues before the movement goal is specified, but representing movement plans once the reach goal has been specified[50,54–57]. In sum, neuronal responses in PPC can reflect the evolution of a spatially specific motor plan based on sensory information and task instruction. Accordingly, parietal computations go beyond mere transformations of spatial location between different reference frames and are flexibly adapted to the current task or behavioral goal.

With the present study, we attempted to approach tactile-motor transformation in a manner that allows for close comparison to well-established findings on visuo-motor transformation. We mapped the emergence of information related to skin-based and external-spatial codes, as well as to the task rule, during a tactile-motor task. We recorded fMRI while human participants planned and executed hand pointing movements to their feet, which were positioned either uncrossed or crossed. A given tactile stimulus is located on the opposite side of space with uncrossed vs. crossed feet; therefore, this manipulation allowed us to dissociate anatomical and external-spatial coding of tactile targets. Moreover, we employed an anti-pointing task rule, which instructed participants to point to the equivalent location to the one that had been touched, but on the other foot, to differentiate tactile sensory processing from motor-related, sensorimotor transformation. Therefore, all movements had to be derived from tactile cues and were directed towards the own body.

## Results
### Experimental setup and analysis rationale
Figure 1 illustrates the trial design. Each trial was separated into four intervals, beginning with a fixation interval. Participants were then presented with a tactile stimulus on their left or right foot, followed by a delay, forming the touch localization interval (duration: 1-4 TR = 1880–7520 ms). There were two stimulus locations on each foot, located medially and laterally on the back of each foot, approximately two centimeters below the heads of the metatarsal bones. We explicitly instructed participants to plan and execute precise pointing movements. This experimental strategy matches that of previous studies, in which pointing movements were directed to visual stimuli at different spatial locations[58,59]. It discourages participants from making stereotypical responses to one side or the other, as it requires precise spatial motor planning. Participants learned the present trial's task rule via a visual cue only after the touch localization interval: either a right-hand pointing movement toward the remembered stimulus location (pro-pointing) or an anti-pointing movement to the homologous location on the other foot. Thus, the target was always a specific location on the own body. This instruction was followed by another delay, forming the movement planning interval (duration: 1-4 TR = 1880-7520 ms). A cue at the end of this interval prompted movement execution. Because the required movement was specified only after the touch localization interval, the final required movement was unknown during this time interval, and therefore we expect that activation in the touch localization interval is related exclusively to stimulus processing. In contrast, activation in the movement planning interval may be related not only to movement

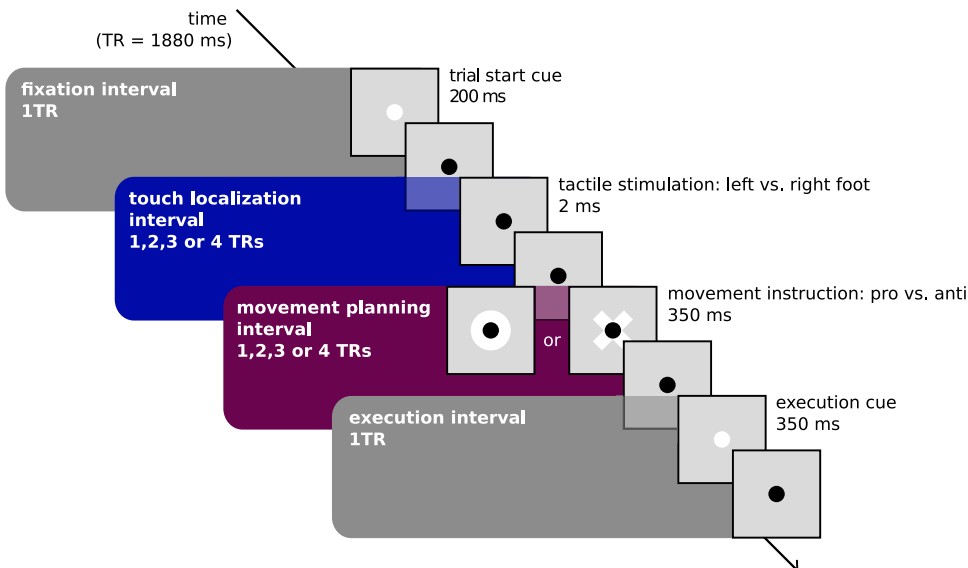

**Fig. 1 | Task design.** A central black fixation dot was present at all times. This fixation dot briefly turned white to indicate the beginning of a trial and was followed by a 1TR (1880 ms) fixed time interval. Next, a tactile stimulus was presented to the left or right foot, followed by delay of variable duration (1-4 TR = 1880-7520 ms). A visual cue then instructed the required movement: a circle instructed pro-pointing of the right hand towards the tactile stimulus; a cross instructed anti-pointing to the homologous location on the other foot. This movement instruction cue was followed by a delay of variable duration (1-4 TR = 1880-7520 ms). Lastly, the fixation dot turned white to prompt movement execution, which was followed by a fixed 1 TR (1880 ms) movement execution interval.

planning, but may also retain spatial information about the stimulus. Participants were highly accurate in pointing movements for trials in which hand movement tracking could be assessed (66% of trials), with a mean accuracy of 92% in making movements to the correct location. A 2 (left vs right tactile stimulation) x 2 (uncrossed vs crossed posture) x 2 (pro vs anti task) within-subjects repeated measures ANOVA showed no main effect of tactile stimulation location ($F_{1,15} = 2.75$, $p = 0.12$, $\eta_p^2 = 0.15$), posture ($F_{1,15} = 0.69$, $p = 0.42$, $\eta_p^2 = 0.04$) or task rule ($F_{1,15} = 0.05$, $p = 0.83$, $\eta_p^2 = 0.00$) on task accuracy, or any interaction between the three factors. Thus, participants were similarly accurate across conditions.

Multi-voxel pattern analysis (MVPA) was performed to assess whether a classifier trained to dissociate fMRI voxel activation patterns observed during one trial class (e.g., trials with a tactile stimulus at the left foot) vs. another trial class (e.g., trials with a tactile stimulus at the right foot) can predict the correct class label of new activation patterns that were not used during training. We used a searchlight procedure[60] that decodes patterns of voxels contained in a sphere with a 4-voxel radius around a center voxel. Each of the brain's voxels serves once as the center voxel, so that the procedure generates a brain-wide map of decoding accuracy. When decoding performance is significantly above chance, the sphere around the respective center voxel contains differences in the patterns of neural responses that can distinguish between the two tested classes. Accordingly, we interpret significant decoding as indicating that the respective region encodes information related to the tested task characteristic.

Classification analyses devised for the present study are summarized in Fig. 2 and specified in detail in Fig. S1. We additionally performed univariate analyses to test whether any regions showed overall differences in activity between our main conditions of interest (see Supplemental Information).

### Anatomical and external spatial coding are present during tactile-sensory processing

We first tested whether information about anatomical and external touch location was present in activity patterns during the touch localization interval, when sensory information was available but the movement goal was not yet specified. Anatomical touch location identifies which of the two feet, right or left, received stimulation, independent of where the respective foot was currently placed. The conditions contrasted in our MVPA decoding are illustrated in Fig. 2A and Fig. S1A (left side, within-interval classifier 1).

Anatomical information was present in multiple parietal regions (Fig. 3A and Table 1): the medial bank of primary somatosensory cortex (S1), bilaterally; the lateral right inferior parietal lobule (IPL), bordering secondary somatosensory cortex (S2) and spreading to lateral S1 and primary motor cortex (M1); and the left anterior SPL. Decoding was also successful outside parietal cortex, in the right PMd and the left insula, spreading into the superior temporal gyrus. The group mean, above-chance decoding accuracy ranged from 53.8–55.0%, with participant-level confidence intervals ranging from 0.3% to 2.0% (Fig. 3B). These classification values are in a range that is typical for MVPA decoding[55,61-65].

External touch location identifies on which side of external space, the right or left side, a stimulus occurred, independent of which foot the stimulus was applied to. Foot crossing dissociates external from anatomical location as, for example, both the uncrossed left foot and the crossed right foot are located on the left side of space. The conditions contrasted in MVPA decoding are illustrated in Fig. 2B and Fig. S1B (left side, within-interval classifier 2). External touch location information was present in a single, right-lateralized cluster confined to the medial IPS (Fig. 4A and Table 1). The group mean, above-chance decoding accuracy was 54.4% with participant-level confidence intervals ranging from 0.4–1.4% (Fig. 4B).

### Tactile-sensory spatial codes are maintained only as long as necessary

Previous research has suggested that neurons change their spatial tuning during sensorimotor processing[53,54]. Therefore, we next tested whether anatomical and external spatial information about the tactile stimulus remained stable across touch localization and movement planning intervals. We used the classifiers trained in the touch localization interval to classify voxel patterns in the movement planning interval[66]. Successful cross-interval classification would suggest that tactile stimulus coding was maintained across both touch localization and movement planning intervals. In

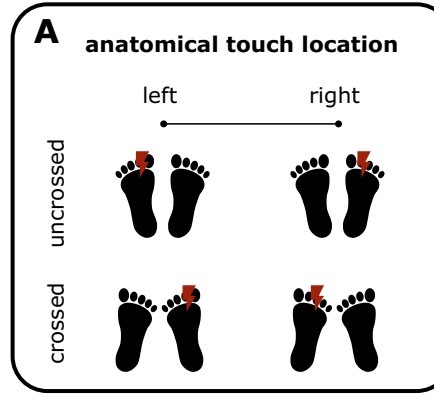

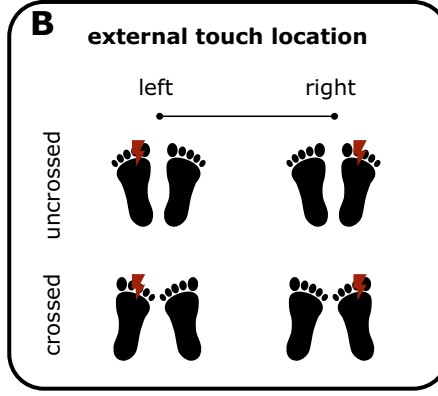

**Fig. 2 | Overview of MVPA decoding analyses. A** Conditions for decoding anatomical touch location (left vs. right foot). **B** Conditions for decoding external touch location, i.e., spatial side of touch. Note, that foot crossing dissociates anatomical from external location. **C** Conditions for decoding the location of the movement goal. The combination of foot posture, stimulation side, and pro/anti-pointing dissociates sensory and movement coding. A more detailed illustration of the MVPA decoding analyses is shown in Fig. S1.

contrast, a failure of cross-interval classification would imply a change in the representational format, as evident in voxel-wise brain activity, across the trial phases.

The conditions we pooled in these analyses are illustrated in Fig. S1 (anatomical location information: Fig. S1A, middle, cross-interval classifier 3; external location information: Fig. S1B, middle, cross-interval classifier 4). These classifiers did not identify any regions in which decoding of touch location across trial phases was above chance. We additionally tested cross-classification of sensory coding separately for pro- and anti-pointing conditions, as it would be possible that a default motor plan could be made in the tactile localization interval towards the target, that would then be remapped for anti-pointing trials upon receiving the task cue. We did not identify any regions that allowed decoding touch location across trial phases for either pro-pointing or anti-pointing trials, suggesting that participants did not prepare a default motor plan to the tactile stimulus during the tactile localization interval. Altogether, the results of these analyses suggest that spatial coding differs between the two trial phases for all regions.

However, even if coding patterns changed from one trial phase to the next, it would still be possible that stimulus-related, anatomical information or external, spatial information is encoded via a different neuronal firing pattern or in different neurons and, thus, expressed in different voxel patterns. To test whether tactile-sensory spatial information is retained during the movement planning interval with an altered coding strategy, we trained classifiers to differentiate anatomical and external target locations with data from the movement planning interval and tested them on their ability to predict the target locations of test data from the same interval (anatomical: Fig. S1A, right side, within-interval classifier 5;

external: Fig. S1B, right side, within-interval classifier 6). Neither classifier identified significant decoding in any region of the cortex. Altogether these classification results indicate that once participants could prepare the movement, sensory information was no longer retained, or was retained at a level too low to be detected by our classification analysis – and, thus, at a level lower than that during the touch localization interval.

In sum, PPC appears to maintain tactile-sensory spatial information only as long as necessary and discard it, or massively reduce its representation, once it can transform sensory information into a motor response. Notably, the two spatial codes were prevalent in distinct PPC regions, without any regional overlap.

### Tactile-motor planning recruits a similar network of fronto-parietal regions as visuo-motor planning

Having established that sensory spatial information is decodable only in the touch localization interval, we next tested which brain regions contain information about the tactually defined spatial location of the movement target during the movement planning interval. The movement goal location identified the location in external space (right or left side) to which a hand pointing movement would be directed; we pooled across all possible stimulus locations on a given foot and decoded only left and right side of space, rather than the exact spatial location. Figure 2C illustrates how the combination of foot posture, stimulus location, and pro vs anti-movement dissociates the location of the movement goal from both anatomical and external stimulus locations in our classification analysis. For example, a pro-pointing movement to the left uncrossed foot, an anti-pointing movement to the left crossed foot, and a pro pointing movement to the right crossed foot all require an identical, left-directed pointing movement but do

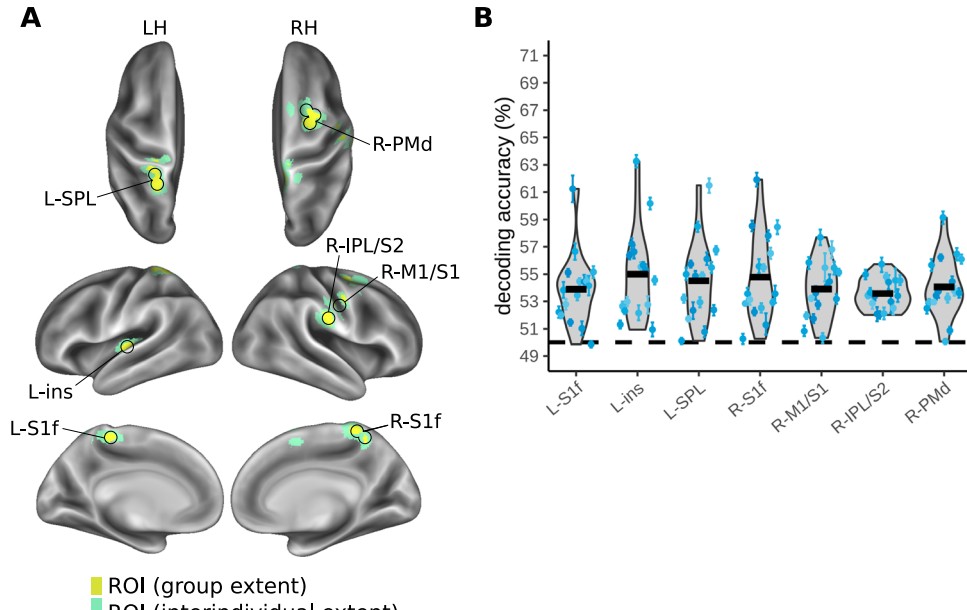

**Fig. 3 | MVPA results for decoding anatomical touch location (right vs. left foot) during the touch localization interval.** An overview of the conditions organized for decoding anatomical touch location is shown in Fig. 2A. **A** Regions of interest (ROI) identified based on a group-level statistical map. The yellow color indicates the extent of the group-level activation (one-sided *t*-test), corrected for multiple comparisons using a cluster-based permutation test (FWE, *p* < 0.05). The green color indicates the spread of the ROI based on region identification for each individual (see Methods). **B** Violin plots illustrate variability of individual decoding accuracy across participants for each ROI (grey shading, black outlines). The dashed black line indicates chance decoding (50%). The thick black horizontal bars indicate the participant group's grand mean decoding accuracy (*n* = 16

participants). The blue scatter points indicate individual means for each participant and the error bars around these individual means represent individual bootstrapped 95% confidence intervals of decoding accuracy. Axis limits correspond with those of Figs. 4, 5, 6 to facilitate comparison across analyses. Source data for Fig. 3B are provided as a Source Data file. LH, left hemisphere; RH, right hemisphere. L-S1f, foot area of left primary somatosensory cortex; L-ins, left insula; L-SPL, left superior parietal lobule; R-S1f, foot area of right primary somatosensory area; R-M1/S1, right primary motor cortex/ primary somatosensory cortex; R-IPL/S2, right inferior parietal lobule/ secondary somatosensory cortex, R-PMd, right dorsal premotor cortex.

---

not consistently share anatomical or external stimulus locations, nor their task rule.

MVPA classification of movement goal location during the movement planning interval (right vs. left movement target, Fig. S1C, within-interval classifier 7) identified widespread above-chance classification in parietal areas, including bilateral S1, M1, and SPL, as well as bilateral frontal areas PMd and SMA (Fig. 5 and Table 1). Above-chance classification was identified in more extensive regions in the left hemisphere, the hemisphere contralateral to the effector that executed the pointing movement. Here, above-chance classification was also evident in parietal cortex along the IPS and in the parietal occipital sulcus (POS), and in frontal cortex in the pre-SMA. In the right hemisphere, a small additional cluster was identified in the occipito-temporal cortex. The group mean above-chance decoding accuracy ranged from 54.1–58.4% in the left hemisphere and from 53.8–58.0% in the right hemisphere, with participant-level confidence intervals ranging from 0.3–2.4% and from 0.3–2.6% respectively (Fig. 5B). In a univariate analysis that identified regions more active for right than left pointing targets, we identified clusters in the bilateral SMA and the left lateral and medial anterior SPL, M1 and the PMd that showed stronger activation for planning pointing movements to right targets as compared to left targets (Fig. S3).

The vast, bilateral responses related to the goal location during the movement planning interval differed markedly from the more regionally confined responses related to sensory spatial information. This is consistent with previous reports of stronger PPC activation during motor planning than sensory processing in sensorimotor delay paradigms such as the current one[58,67], underlining the prominent involvement of PPC in motor planning and control[49,68].

## PPC encodes the current task rule

Our task required participants to interpret the task rule, i.e., pro- vs. anti-pointing, to derive the required movement from the tactile location. Participants showed similar mean task accuracy for both pro- and anti-pointing (both 92% correct). In the macaque PPC, the task rule is encoded independent of specific sensory cues[69], and it modulates neuronal firing in the areas that encode the movement goal[49]. Human fMRI studies did not find differences in univariate activation between preparation of pro- and anti-pointing movements[50,54], nor were they able to decode pro- vs. anti-pointing from regions of interest in the SPL, aIPS or PMd[55]. We tested whether we could decode the task rule in our tactile-motor task. Figure 6 illustrates the performance of a classifier trained to dissociate pro- and anti-pointing movements during the movement planning interval (Fig. S2, within-interval classifier 8). Average ROI coordinates across participants are displayed in Table 1. Task rule could be decoded above chance-level bilaterally in the SPL and in the left superior parieto-occipital cortex (SPOC). The mean above-chance decoding accuracy of the task rule per ROI ranged from 53.8% to 54.1% with participant-level confidence intervals covering ranges from 0.3% to 1.3% (Fig. 6B).

## From touch localization to sensorimotor planning: functional overlap

Our results demonstrate that PPC contains anatomically and externally coded tactile location information during the touch localization interval and information about the motor goal and task rule during the movement planning interval. We next explored whether there was overlap between the regions that encoded sensory and motor-related spatial information in the two intervals. Such an overlap would provide

**Table 1 | Means and standard errors of group ROIs in MNI coordinates, grouped by classification analysis (anatomical touch location; external touch location; movement goal location, movement instruction)**

| Analysis | Hemisphere-ROI | Mean | | | Standard Error | | |
|---|---|---|---|---|---|---|---|
| | | x | y | z | x | y | z |
| Anatomical touch location | L-S1f | −5.6 | −37.4 | 60.3 | 0.7 | 0.9 | 0.5 |
| | L-ins | −35.8 | −23.9 | 5.9 | 0.7 | 0.8 | 0.8 |
| | L-SPL | −21.1 | −43.3 | 64.2 | 0.4 | 0.3 | 0.7 |
| | R-S1f | 8.4 | −42.2 | 63.1 | 0.5 | 0.4 | 0.4 |
| | R-M1/S1 | 58.7 | −16.9 | 28.4 | 0.7 | 0.9 | 0.6 |
| | R-IPL/S2 | 50.6 | −10.9 | 41.0 | 0.7 | 0.9 | 0.5 |
| | R-PMd | 23.9 | −0.3 | 50.6 | 0.5 | 0.5 | 0.3 |
| External touch location | R-mIPS | 24.8 | −59.2 | 51.3 | 0.5 | 0.5 | 0.6 |
| Movement goal location | L-M1h | −31.7 | −21.6 | 63.9 | 0.7 | 0.9 | 0.5 |
| | L-S1h | −30.8 | −29.3 | 50.8 | 0.6 | 0.9 | 0.7 |
| | L-S1f | −3.6 | −45.8 | 57.1 | 0.9 | 0.6 | 0.8 |
| | L-PMd | −26.4 | −7.8 | 60.5 | 0.6 | 0.8 | 0.8 |
| | L-SMA | −7.9 | −6.8 | 55.4 | 0.7 | 0.8 | 0.8 |
| | L-preSMA | −10.1 | −6.6 | 46.6 | 0.7 | 0.7 | 0.8 |
| | L-aIPS | −34.9 | −35.8 | 45.9 | 0.4 | 0.7 | 0.6 |
| | L-SPL | −16.7 | −53.3 | 59.0 | 0.4 | 0.3 | 0.4 |
| | L-SPOC | −14.9 | −79.9 | 37.2 | 0.5 | 0.4 | 0.6 |
| | R-M1h | 33.0 | −21.8 | 64.4 | 0.6 | 1.1 | 0.5 |
| | R-S1h | 29.6 | −30.4 | 51.3 | 0.7 | 0.8 | 0.6 |
| | R-S1f | 4.3 | −45.4 | 56.2 | 0.8 | 0.6 | 0.7 |
| | R-PMd | 26.6 | −7.9 | 62.0 | 0.6 | 0.9 | 0.7 |
| | R-SMA | 6.0 | −6.6 | 55.6 | 0.5 | 0.8 | 0.7 |
| | R-SPL | 13.1 | −55.3 | 56.5 | 0.4 | 0.4 | 0.5 |
| | R-V5 | 39.0 | −65.7 | 17.2 | 0.7 | 0.7 | 0.8 |
| Movement instruction | L-SPL | −14.5 | −58.2 | 58.3 | 0.2 | 0.5 | 0.4 |
| | L-SPOC | −15.2 | −77.5 | 46.6 | 0.8 | 0.5 | 0.7 |
| | R-SPL | 5.5 | −59.4 | 55.1 | 0.7 | 0.5 | 0.5 |

*L* left hemisphere, *R* right hemisphere. *S1f* foot area of primary somatosensory cortex, *ins* insula, *SPL* superior parietal lobule, *S1f* foot area of primary somatosensory cortex, *M1/S1* primary motor cortex/ primary somatosensory cortex, *IPL/S2* inferior parietal lobule/ secondary somatosensory cortex, *PMd* dorsal premotor cortex, *mIPS* medial intraparietal sulcus, *M1h* hand area of primary motor cortex, *S1h* hand area of primary somatosensory cortex, *SMA* supplementary motor area, *preSMA* pre-supplementary motor area, *aIPS* anterior intraparietal sulcus, *SPOC* superior parieto-occipital cortex, *V5* middle temporal visual area.

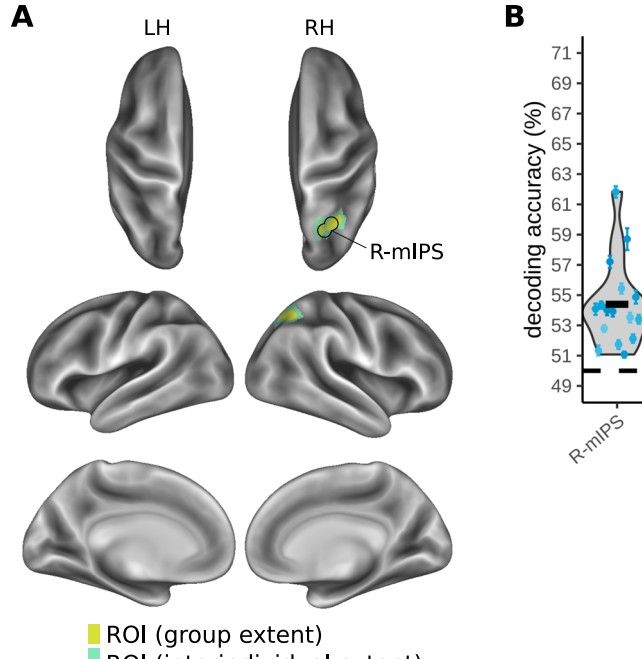

**Fig. 4 | MVPA results for decoding external touch location (right vs. left side of space) during the touch localization interval.** An overview of the conditions organized for decoding external touch location is shown in Fig. 2B. **A** Right medial intraparietal sulcus (IPS) ROI (R-mIPS) identified based on a group-level statistical map. The yellow color indicates the extent of group-level activation (one-sided t-test), corrected for multiple comparisons using a cluster-based permutation test (FWE, *p* < 0.05). The green color indicates the spread of the ROI based on region identification for each individual (see Methods) **B** Variability of individual decoding accuracy across participants in R-mIPS. The thick black horizontal bars indicate the participant group's grand mean decoding accuracy (*n* = 16 participants). The blue scatter points indicate individual means for each participant and the error bars around these individual means represent individual bootstrapped 95% confidence intervals of decoding accuracy. Source data for Fig. 4B are provided as a Source Data file.

evidence for a dynamic change in spatial coding within areas across the duration of a trial, likely implying that the same regions are involved in the underlying transformation between the different spatial codes over the course of sensorimotor processing.

Figure 7 displays the overlap between clusters in which our classifiers identified sensory and motor-related spatial coding. There is considerable overlap between these spatial codes across trial intervals. 34% of the voxels that encoded the anatomical stimulus location during the stimulus localization interval overlapped with voxels identified as coding movement goal location in the movement planning interval. These overlapping voxels were located in S1 bilaterally, as well as the left SPL and the right PMd (Fig. 7A; overlap: 221 voxels = 5967 mm³). 61% of the voxels that encoded external stimulus location during the stimulus localization interval overlapped with voxels that coded movement goal location. These voxels were located in the right mIPS (Fig. 7B; overlap: 74 voxels = 1998 mm³). Thus, a considerable proportion of voxels participated in coding spatial stimulus information in one reference frame in the first trial interval, and

spatial movement goal information in a different reference frame in the second trial interval.

Furthermore, 90.4% of the cluster representing the task rule overlapped with the cluster representing movement goal locations bilaterally in the SPL and in the left POS (overlap of 422 voxels, volume: 11,394 mm³; Fig. 7A). In the left SPL, located between voxels common to anatomical and movement goal location and voxels common to goal location and task rule, was a voxel overlap of all three types of information (overlap of 34 voxels, volume: 1161 mm³; Fig. 7A). This finding implies that the prevalent information coded in these regions, as decodable with MVPA, varies over the course of the trial and, thus, that the neural function of these regions changes over time: PPC regions that coded tactile location before a movement plan could represent the movement goal once the task rule had been specified. Moreover, areas that carried information about the task rule were almost completely enclosed within the cluster that coded movement goal location, suggesting that this bilateral PPC region performs a sensorimotor transformation based on stimulus location and task rule, to turn this information into a movement goal.

The results of decoding anatomical touch location, external touch location, movement goal location and task rule presented here were robust against variations of the analysis pipeline and model specification. We re-ran separate instances of the presented analyses to address several aspects that could potentially bias our results (see Supplementary Information). First, we employed an unwarping

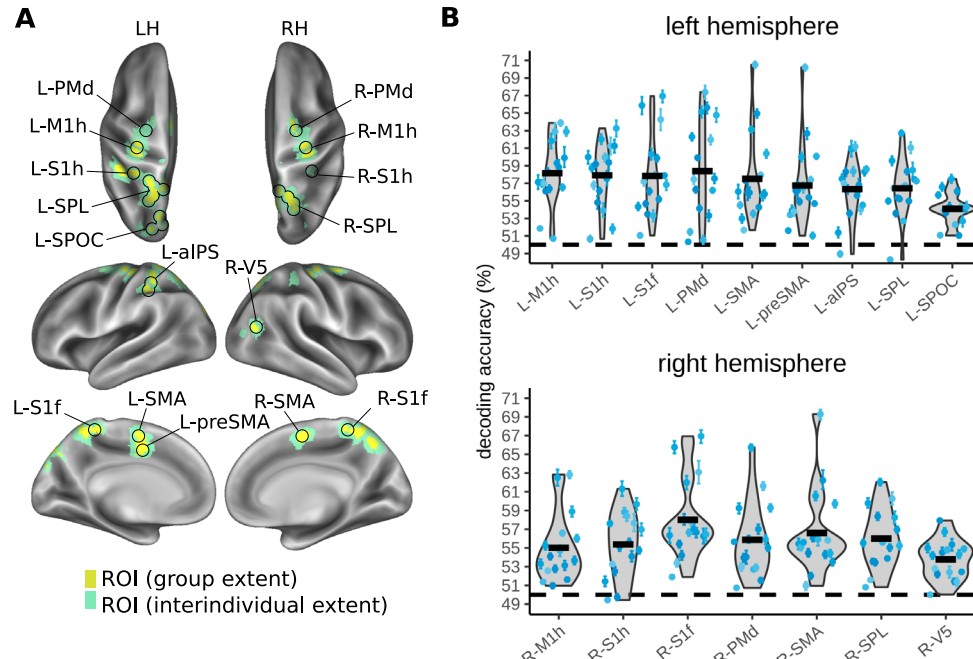

**Fig. 5 | MVPA results for decoding movement goal location (pointing movement to the foot on the right or left side of space) during the movement planning interval.** An overview of the conditions organized for decoding movement goal location is shown in Fig. 2C. **A** ROIs identified based on a group-level statistical map. The yellow color indicates the extent of group-level activation (one-sided *t*-test), corrected for multiple comparisons using a cluster-based permutation test (FWE, *p* < 0.05). The green color indicates the spread of the ROI based on region identification for each individual. **B** Variability of individual decoding accuracy across participants in the different ROIs. The thick black horizontal bars indicate the participant group's grand mean decoding accuracy (*n* = 16 participants). The blue scatter points indicate individual means for each participant and

the error bars around these individual means represent individual bootstrapped 95% confidence intervals of decoding accuracy. Source data for Fig. 5B are provided as a Source Data file. L-M1h, hand area of left primary motor cortex; L-S1h, hand area of left primary somatosensory cortex; L-PMd, left dorsal premotor cortex; L-SMA, left supplementary motor area; L-preSMA, left pre-supplementary motor area; L-aIPS, left anterior intraparietal sulcus; LSPL, left superior parietal lobule; L-SPOC, left superior parieto-occipital cortex; R-M1h, hand area of right primary motor cortex; R-S1h, hand area of right primary somatosensory cortex; R-PMd, right dorsal premotor cortex; R-SMA, right supplementary motor area; R-SPL, right superior parietal lobule; R-V5, right middle temporal visual area.

procedure that included removal of motion artifacts and, therefore, did not include motion regressors in our GLMs. Re-running our analyses with motion regressors included did not result in any notable differences to the results we report here (see Fig. S4). Second, we had included trials in our analyses even if we had been unable to extract the finger movement response, for example due to knee or leg posture obstructing the view of the hand. Re-running our analyses without such trials rendered very similar results to the ones we report here (Figure S5). Third, we had modelled each trial phase for its true duration of 1-4 TRs. While common for delayed-movement paradigms such as the one we use here[50,58,67,70,71], this procedure favors sustained activity over transient responses. Re-running our analyses with a 1-TR duration for each trial phase revealed mostly comparable results as our main analysis (see Figs. S6 & S7). A notable difference, however, was that external touch location in mIPS was evident bilaterally rather than unilaterally.

## Discussion

We investigated the reference frames involved in tactile-motor transformation during pointing to tactually indicated locations on the feet. We dissociated anatomical and external-spatial coding of tactile stimulus location by manipulating whether participants' feet were uncrossed or crossed, and further dissociated between sensory and motor-related responses using a delayed anti-task paradigm. We report three key findings. First, PPC exhibited concurrent anatomical and external-spatial coding of tactile stimulus location in different PPC regions during the touch localization interval. Anterior regions including primary somatosensory cortex and SPL encoded stimulus location in an anatomical reference frame, whereas medial IPS, located

in the posterior PPC, encoded stimulus location in an external spatial reference frame. Second, spatial coding was dynamic, showing responses related to stimulus location prior to presentation of the task rule, and then responses encoding the motor goal location following presentation of the task rule. There was overlap between regions encoding first sensory and later motor information, and one region in SPL coded the original anatomical stimulus location, the task rule, and the resulting movement goal. Thus, this region may be central in transforming tactile information into an actionable spatial target. Third, coding of the movement goal was present in a large network, consistent with regions shown to be involved in motor responses to visual targets, suggesting similar coding of movements to tactually and visually defined targets.

**Anterior PPC encodes touch in an anatomical reference frame**

We identified regions that use an anatomical code for tactile location by testing which regions could decode the foot on which the tactile stimulus was received, regardless of crossed or uncrossed foot posture. Decoding was above chance in the SPL, a region in the anterior PPC, as well as in a frontoparietal network of regions known to be involved in touch processing. Previous work has demonstrated that anterior PPC contains regions that show selective responses to tactile stimulation on different body parts[26,31,32,72]. In particular, a human fMRI study identified a roughly homuncular tactile map in the SPL and anterior IPS that also overlapped with retinotopic visual maps[32,72]. Tactile leg and toe regions in this map lay medially, in close agreement with the SPL cluster for anatomical foot location found in the present study (Fig. 8: yellow sphere number 1). Here, we report that this region uses an anatomical code

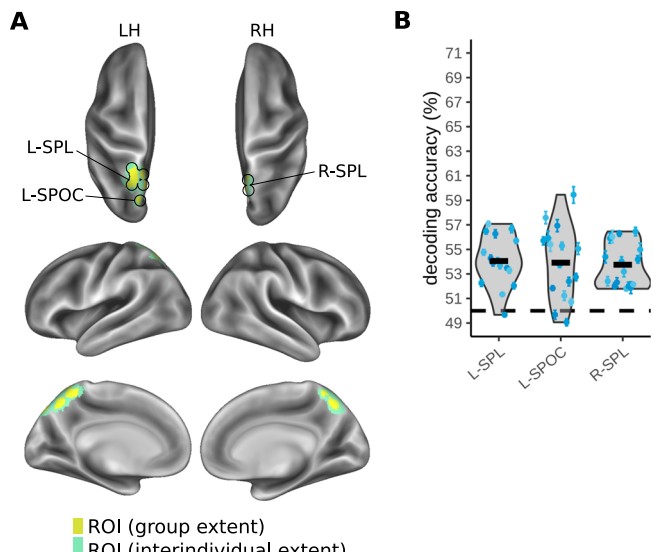

**Fig. 6 | MVPA results for decoding task rule (pro- vs. anti-pointing movement) during the movement planning interval. A** ROIs identified based on a group-level statistical map. The yellow color indicates the extent of group-level activation (one-sided t-test), corrected for multiple comparisons using a cluster-based permutation test (FWE, p < 0.05). The green color indicates the spread of the ROI based on region identification for each individual. **B** Variability of individual decoding accuracy across participants in the different ROIs. The thick black horizontal bars indicate the participant group's grand mean decoding accuracy (n = 16 participants). The blue scatter points indicate individual means for each participant and the error bars around these individual means represent individual bootstrapped 95% confidence intervals of decoding accuracy. Source data for Fig. 6B are provided as a Source Data file. L-SPL, left superior parietal lobule; L-SPOC, left superior parieto-occipital cortex; R-SPL, right superior parietal lobule.

for tactile stimuli, that is, location is coded regardless of the current position of the limb in external space. This anatomical stimulus coding in anterior PPC may not be limited to tactile sensation. Human fMRI studies have identified coding of visual stimuli relative to the position of the body in anterior PPC[19,20,23] (Fig. 8: cyan spheres), suggesting that anterior PPC may encode multisensory stimuli in relation to the body's layout or to the skin. This finding extends the suggestion that rostral PPC "projects" the environment onto the body and estimates the current state of the environment by transforming information to match with the own body[26,73].

We decoded anatomical tactile stimulus location bilaterally in the medial bank of bilateral S1, adjacent to the SPL. This is consistent with the role of this region as the part of the somatosensory homunculus that responds to contralateral foot stimulation[1,2,74]. Regions beyond S1 also exhibited sensitivity to the anatomical location of tactile stimuli, including the left insula and the right IPL/S2, M1/S1 and PMd. Previous studies also reported that these regions respond to tactile stimuli[75–77], but did not specify the reference frame used to encode these stimuli. Viewed together, these previous and our present results identify a network of brain regions that encode tactile sensory location in an anatomical reference frame.

### Posterior PPC encodes tactile location in an external-spatial reference frame

We identified a region in the mIPS that encoded tactile stimulus location in an external-spatial reference frame. This coding implies that the location has been abstracted from the skin surface: a given location is coded relative to where it is in space, no matter on which body part or skin location it was originally received. This external-spatial reference frame may be gaze-, head-, trunk- or hand-centered. We did not target these different possibilities in our study, and accordingly did not vary

gaze, head, body and hand position but instead kept all four aligned in our experimental setup.

Recent experiments have cast some doubt on the idea that touch is truly recoded into an external-spatial code[78,79]. These studies point to a dissociation between the assignment of a tactile stimulus to the limb on which it occurred and its 3D external-spatial location. They suggest that touch is automatically associated with the space that the touched limb usually resides in – such as the right side of the body for the right arm and hand – and that the true external-spatial location of the touch is not derived automatically, but only when required. This conceptual difference may appear subtle: whether a tactile stimulus is directly associated with a location in 3D space, or whether it is first assigned to a body part and that body part is then localized in space, it is always posture information that must be integrated with the skin location of the tactile stimulus. Yet, the second view implies an important difference between visual and tactile processing, because it allows for the possibility that the tactile stimulus itself is never coded in 3D space. A movement towards a touch may, instead, involve referring the touch to a limb, identifying where the limb is in space, and then planning a movement to a location on the limb[80].

The activated mIPS region identified in the present study partially overlapped with a region previously reported as a putative human homologue of the lateral intraparietal area (LIP), which has been associated with eye-centered coding[15,56,81–83] (Fig. 8: green spheres). LIP encodes both visual and auditory targets in an eye-centered reference frame[84–86] and it was therefore proposed that LIP generally encodes stimuli of all modalities in an eye-centered reference frame[13]. Our results are consistent with this proposal. Whereas LIP is often considered to be specific to saccade planning, it has also been suggested to provide a sensory priority map[87,88]. Furthermore, LIP neurons are active during reaching movements[87,89]. This finding fits with reports that putative human homologue of LIP responds to both saccades and reaches[15,58,59,90,91]. Thus, a contribution of LIP to our present pointing task is consistent with previous findings, especially given that we observed partial overlap between mIPS voxels initially coding tactile stimuli in an external-spatial reference frame and later responding to the movement goal. Overall, LIP's role in our delay task may be to first maintain salient spatial target information and later to transform it into a motor goal once participants receive the pro/anti task rule.

However, the mIPS region we identified in our study also partially overlapped with hVIP#1, a region that we have proposed to be one of three areas that together form the human homologue of macaque VIP[26]. Previous findings suggest that hVIP#1 encodes the location of sensory stimuli in an external-spatial reference frame[92], which again fits well with our present results. In sum, mIPS appears to encode tactile information projected into an external-spatial code that is independent of the skin and is likely anchored to the eyes.

Further support for a role of the LIP/ hVIP#1 region identified here in external-spatial coding of touch location has come from several TMS studies. Figure 8 depicts the respective stimulation locations; three sites fall within hVIP#1[38,39,93], and one site lies more posteriorly, close to the coordinates of the putative human LIP[37]. TMS to these locations impaired participants in making judgments that required transforming tactile location from the skin into space[37], appeared to impede, or modulate, the integration of arm posture in the context of tactile-spatial processing[38,39], and impaired the integration of auditory cues with tactile processing in external-space[93]. Despite their locational variability, a common aspect of all these studies was that TMS resulted in tactile stimuli being processed as if they had been assigned to the correct limb but had not been referred to the location in space at which that limb currently lay but, instead, to where that limb is normally located. These findings have typically been interpreted as indicating that posterior PPC plays a role in remapping tactile stimuli from an anatomical to and external-spatial reference frame. However, they are also in line with the idea we introduced earlier, namely that touch is

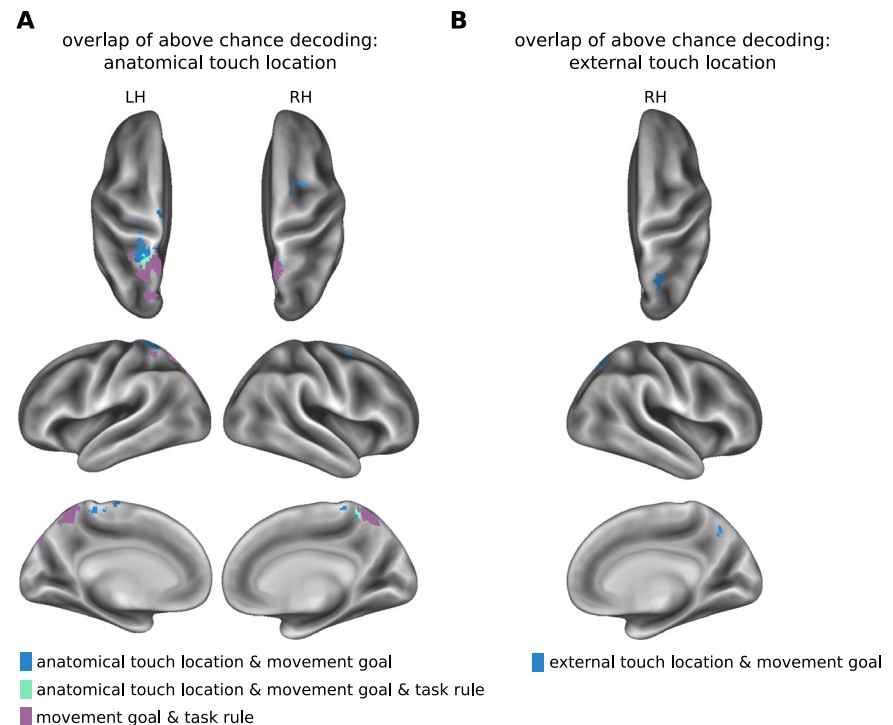

**A** overlap of above chance decoding: anatomical touch location

**B** overlap of above chance decoding: external touch location

■ anatomical touch location & movement goal
■ anatomical touch location & movement goal & task rule
■ movement goal & task rule

■ external touch location & movement goal

**Fig. 7 | Overlap of clusters encoding information about anatomical or external touch location, movement goal location, and task rule. A** Overlap with regions that encode anatomical touch location. **B** Overlap with regions that encode external touch location. LH, left hemisphere; RH, right hemisphere.

referred to a limb and it is the location of that limb that is tracked in space, rather than the touch itself.

Decoding touch in space was only possible in the right hemisphere in our study. This finding is consistent with the proposal that right PPC may be specialized for spatial somatosensory function, as somatosensory deficits related to spatial processing are more frequent following lesions in the right PPC as compared to the left PPC[94,95]. However, it appears inconsistent with the widely held idea that each hemisphere of PPC represents mainly the contralateral side of space, and findings that both putative human LIP and hVIP#1 are bilateral regions[15,26,56,81,83]. The solution to this apparent contradiction may be resolved by the results of our re-analysis that was geared towards transient, short-term responses by modelling only 1 TR after tactile and task rule cues. This analysis decoded external-spatial touch in bilateral posterior PPC (see Figure S6). This result suggests that LIP/hVIP#1 use a distributed spatial code for sensory processing, but that the right hemisphere carries some specialized functions with regard to rule-based integration of this sensory location, which is caught by the interval-long predictors in our analyses. This reasoning may also explain why fMRI responses related to tactile remapping in PPC were stronger in the right hemisphere in one study[96], but in the left hemisphere in another study[97]. TMS studies on tactile remapping have often targeted only the right hemisphere, making it unclear whether their findings would generalize to the left hemisphere[37,38,93]; however one study that stimulated both the left and right hemisphere found similar effects for both hemispheres[39]. In sum, the available evidence across fMRI and TMS studies favors the view that spatial representation in posterior PPC is distributed and bilateral, but that lateralization exists for some higher-order, rule-based processes.

**Two possible views of how touch is remapped**

The overall picture that emerges is that anterior and posterior PPC regions play opposite roles in coding tactile stimulus location. Anterior regions relate the environment to the body and, thus, code space anchored to individual body parts. In contrast, posterior regions relate the body to the environment and code information in

an external-spatial reference frame, potentially anchored to the direction of gaze (see ref. 73). Thus, we concurrently decoded touch location in anatomical and external-spatial reference frames across PPC. Modelling[98], behavioural[99–101], electrophysiological[35,36,102] and magnetoencephalographic[34,43] studies have demonstrated that tactile information is encoded in multiple, concurrently active, reference frames. Extending these previous findings, our study disentangled the respective involved, parietal brain regions at high spatial resolution.

Our study was not designed to resolve whether it is touch itself, or the touched limb, which is localized in 3D space for the pointing response. The presence of the external spatial location coding in the sensory phase supports the view that the tactile stimulus itself was coded in space. Yet, the present task instructions explicitly required the use of the external-spatial location of touch. Therefore, representation of tactile location already during the sensory trial phase, in which movement was not specified, may have been induced by the general task requirements and may not have occurred in the absence of such a task. In this case, the external information we decoded could be related directly to the external location of the tactile stimulus, or to the location of the limb to which the tactile stimulus was assigned. PPC responds to changes in body posture[40–42,103], suggesting that PPC probably also encodes information about current body posture. We did not directly address the coding of body posture – crossed vs. uncrossed feet – in the present study. This is because we manipulated posture across runs, so that any differences between postures identified by MVPA could be attributable to unknown differences between runs, rather than limb crossing per se.

Finally, it is conceivable that participants could have developed a default plan for a pro-pointing movement during the stimulus presentation phase of the trial, that is, before they knew whether they would have to pro- or anti-point. Such a default plan could be maintained both for a recoded tactile location and for a derived target limb. It has been proposed that movement choice tasks could involve planning for one of the available target options during a delay period, and that this default plan could then be switched if the final cue

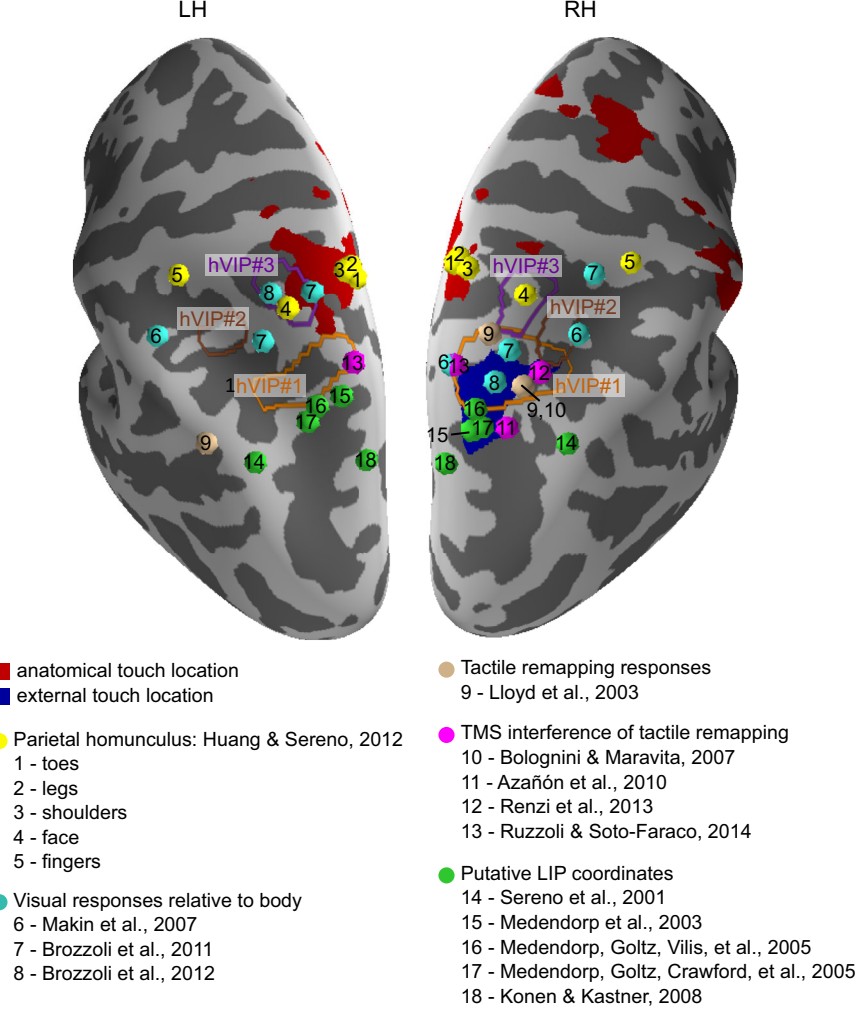

LH                                                         RH

anatomical touch location
external touch location

Parietal homunculus: Huang & Sereno, 2012
1 - toes
2 - legs
3 - shoulders
4 - face
5 - fingers

Visual responses relative to body
6 - Makin et al., 2007
7 - Brozzoli et al., 2011
8 - Brozzoli et al., 2012

Tactile remapping responses
9 - Lloyd et al., 2003

TMS interference of tactile remapping
10 - Bolognini & Maravita, 2007
11 - Azañón et al., 2010
12 - Renzi et al., 2013
13 - Ruzzoli & Soto-Faraco, 2014

Putative LIP coordinates
14 - Sereno et al., 2001
15 - Medendorp et al., 2003
16 - Medendorp, Goltz, Vilis, et al., 2005
17 - Medendorp, Goltz, Crawford, et al., 2005
18 - Konen & Kastner, 2008

**Fig. 8 | Comparison of anatomical and external-spatial coding of tactile stimuli to coordinates from previous studies.** Anatomical coding of touch location is overlayed in red and external-spatial coding of touch location is overlayed in blue. Outlines of the human ventral intraparietal (hVIP) complex[26] are shown with hVIP#1 outlined in orange, hVIP#2 outlined in brown and hVIP#3 outlined in purple. Coordinates of the locations of body parts identified as a parietal homunculus[32] are shown in yellow. Coordinates of visual responses relative to the position of the body are shown in cyan. Coordinates of responses related to tactile remapping are shown in light brown. Locations of coordinates that were reported to interfere with tactile remapping when stimulated using transcranial magnetic stimulation are shown in pink. Locations of the putative coordinates of the human homologue of the lateral intraparietal area (LIP) are shown in green.

requires using the non-prepared movement[104]. Nevertheless, we deem it unlikely that a default motor plan accounts for our present results. The activation during the movement planning phase of trials was much more extensive than that during the sensory phase, during which a default plan should have been evident if it existed. Furthermore, we found no evidence that coding was maintained across sensory target localization and motor planning intervals. Recordings from neurons in macaque LIP also support this notion: macaques delayed forming their decision until the final movement plan could be made, even when sensory decision information was already available[105]. Thus, altogether PPC appears to flexibly change its coding from sensory processing to motor preparation.

**Goal-directed pointing recruits similar networks independent of target modality**
Once both target and task rule had been defined and participants could plan the required action, a large network spanning multiple regions in premotor cortex, primary sensorimotor cortex, and PPC selectively represented spatial information about the movement goal. This network closely corresponded to the frontoparietal network involved in visuo-motor planning[59,106–111]. Although regions of both

hemispheres contained movement planning information, a more extensive area of the left, than right, hemisphere could decode the movement goal, and decoding accuracy was also higher in the left hemisphere. This is likely because participants pointed with their right hand. Regions coding the movement goal location included SPOC, the putative human homologue of the parietal reach region, and PMd; these two regions are thought to be involved in calculating the movement path from effector to target location[67,91,112,113]. The movement planning network also spanned the M1 hand area, SMA and pre-SMA, which are known to code preparatory movement signals related to the effector and target, as well as object-related movement intentions[61,63,114–116]. Lastly, the movement planning network also encompassed an anterior IPS region that has been linked to the preparation of grasping and pointing movements[10,58,71,117,118]. Altogether, these results identify a broad network of regions that encode the tactually-defined movement goal location and suggest that movement planning is very similar across sensory modalities of the target stimuli.

In natural behavior, movements in response to tactile stimuli often involve directly touching the stimulated location or limb. For example, one might brush the right hand across the left arm after feeling an insect crawling along it. Our study differs from such natural

situations and behavior in some regards. First, in our study, participants pointed towards, rather than moved to, the locations indicated by touch. fMRI work has compared responses for pointing and reaching and observed similar activation for both in a frontoparietal network consistent with the regions in which we decoded the pointing movement goal in the present study[119]. Notably, touching the own skin gives rise to additional somatosensory feedback that has been shown to improve tactile localization[120], and there can be attenuation of somatosensory signals due to predictive mechanisms[121,122]. Second, targeting the own body with a movement often involves moving not only the acting limb, but also the target area. In the above example, the right hand (actor) and the left arm (target) would probably both move to meet in front of the torso–an area primates prefer for both manual manipulation and viewing of objects[123]. These considerations show important differences between visuo-motor and tactile-motor planning and highlight exciting areas for future research. However, our aim with the present study was to use an experimental design that was as similar as possible to visuo-motor research paradigms. In such paradigms, participants can only move the acting effector, but not the target, and experiments have used pointing and reaching responses alike to explore the transformation from a sensory to motor code.

### Sensory representations are only maintained as long as necessary

We were unable to decode sensory location information in the part of the trial in which the task rule had been specified and participants were, thus, able to plan the pointing movement. Classifiers trained to decode sensory location during the touch localization interval did not generalize to the movement planning interval, nor was decoding successful when we attempted to decode sensory location by both training and testing classifiers on responses during the movement planning interval. These findings suggest that sensory representations were only maintained for as long as necessary. Compatibly, previous fMRI work investigating visuo-motor control has also demonstrated that sensory codes are only maintained as long as they are relevant in most regions[54,55], and recordings from PRR neurons in macaques have shown only brief, transient encoding of the visual target location when the movement cue is already known[49]. Our task required a movement of the effector to a stationary tactile target, which may have aided the loss of sensory coding once the movement could be planned. Different results may be obtained when either the tactile target moves along the body, or when participants execute a movement that involves both the acting and target limbs, as discussed above. However, such considerations are in line with an interpretation that sensory codes are maintained as needed.

In the present study, brain regions that first encoded sensory information overlapped with those that later encoded movement planning (Fig. 7). This finding suggests that a dynamic sensorimotor transformation occurred in these regions once information to plan the pointing movement was available. Such overlap of target location and motor goal location across time has also been identified in the PMd and posterior IPS during visuo-motor tasks[54], and equivalent dynamics have been identified in the coding of neurons in the macaque parietal reach region[49].

### Posterior parietal cortex encodes the movement task rule

We decoded the movement task rule (i.e., pro- vs. anti-pointing movement) from bilateral PPC clusters in the SPL and SPOC. The success of previous human fMRI studies that employed an anti-task visuo-motor paradigm to decode the task rule from PPC have been mixed. Some studies did not observe differences in univariate activation evoked during planning of pro- and anti-movements[50,54], whereas another study reported regional differences for pro- and anti-pointing movements[48]. Yet another study was unable to identify regions representing the task rule using a decoding approach[55], but used a

region of interest approach and may therefore have overlooked task rule encoding in areas outside the inspected regions. In contrast to these human fMRI studies, responses of neurons in monkey PRR were modulated by the pro/anti task rule[49], and PPC neurons encoded task rule information independent of specific sensory cues[69]. In our study, the bilateral SPL and SPOC regions that encoded the task rule almost completely overlapped with regions that encoded the movement goal. The respective left SPL cluster additionally overlapped with the SPL cluster that had previously encoded the anatomical tactile target location. These findings suggest that PPC regions simultaneously encode multiple features at the population level[124].

It is of note that the ability to relate sensory cues to arbitrary rules links to the above-discussed framework of limb identification as an intermediate step to tactile-motor interaction. One could view the tactile stimulus as a cue that instructs a movement towards a body part. In this view, tactile-motor transformation is indirect, with the connecting "rule" being a mapping between a body map and space mediated by current body posture.

### Dynamic spatial transformations in posterior parietal cortex mediate tactile-motor transformation

Our study reveals the dynamic representation of spatial information during tactile sensorimotor planning. Spatial codes in PPC were characterized by distinct multivariate activation patterns that changed their selectivity from representing sensory information during target localization to representing the movement goal during motor planning. Tactile-spatial sensory information was concurrently encoded in anatomical and external-spatial reference frames. These different codes were evident in distinct PPC locations along an anterior to posterior gradient, consistent with recent proposals that PPC contains poles that relate the environment to the body (anterior) and vice versa (posterior)[73]. Once the movement task rule was specified, tactile information was no longer detectable, and instead a frontoparietal network encoded the location of the movement goal. Information about the tactile stimulus, movement goal and the movement task rule converged in the left SPL, suggesting this region may play a key role within the network that transforms tactile location into a movement plan. Overall, the posterior parietal cortex integrates sensory information of the different sensory modalities provided in different reference frames and flexibly combines this information to plan movement towards both the environment and the own body.

## Methods
### Participants

The analyzed sample consisted of 16 students of the University of Hamburg (10 female, 6 male, determined by self-reporting), mean age 23.8 years (range: 19–30 years). Participants were right-handed according to questionnaire-guided self-report[125], had normal or corrected-to-normal vision, and reported to be free of any neurological disorders, movement restrictions, or tactile sensitivity problems. Participants provided written informed consent and received course credit or € 8/hour for their participation. Four further participants were excluded from analysis: one had performed >99% of all movements to the incorrect movement goal in the crossed, but not in the uncrossed foot posture, MR slice positioning accidentally omitted part of SPL for two participants, and only partial data was collected due to a technical error for one pilot participant.

The experiment was approved by the ethics committee of the German Association of Psychology (Deutsche Gesellschaft für Psychologie, DGPs, TB 102011 and TB 102011_add_092014).

### Behavioral task

Participants planned and executed right hand movements toward tactile stimuli presented on the feet, while blood oxygenation level dependent (BOLD) signal changes in the brain were recorded using

**fMRI.** A delayed movement task separated tactile localization, movement planning and movement execution into separate phases[15,58,59,67,91,126]. Figure 1 illustrates the four phases of each trial: fixation (duration: 1 TR = 1880 ms), tactile localization on the foot (duration: 1-4 TR = 1880-7520 ms, in steps of 1 TR), movement planning of the right index finger (duration: 1-4 TR = 1880-7520 ms, in steps of 1 TR), and movement execution (duration: 1 TR = 1880 ms). The tactile stimulus occurred pseudo-randomly on the back of the left or right foot, either medially or laterally, approximately two centimeters below the heads of the metatarsal bones. Critically, the tactile stimulus was not informative about the movement that would be required later in the trial: at the beginning of the movement planning delay, a visual cue indicated whether the pointing movement should be planned directly toward the tactile stimulus (pro-movement) or toward its mirrored location (anti-movement). It was only at this point in time that participants could derive the full movement plan. Finally, a visual cue prompted execution of the finger pointing movement. Participants were instructed to point precisely but did not receive feedback on their pointing movement accuracy.

In different experimental blocks, participants assumed either an uncrossed or crossed foot posture. The crossed posture allows anatomical and external locations of tactile stimuli to be dissociated as, for example, a stimulus on the anatomically right foot is then located in the left side of space.

The experiment comprised 3 within-subject factors. Foot Posture (levels: uncrossed vs. crossed) was maintained for 3 runs in a row, with the start posture alternated between sessions and counterbalanced across participants. We chose to minimize posture changes during the experiment to avoid fMRI and movement video recording artefacts due to changes in head and hand position. Stimulated Foot (left vs. right foot; collapsed across the medial and lateral stimulus locations on each foot) and Instructed Movement (pro-movement vs. anti-movement) varied from trial to trial. Trials were distributed into blocks of 33 trials each, with additional rest phases of 11 and 4 TR at the beginning and end of each run, respectively. We minimized dependencies between trials by balancing the trial sequence run-wise such that each condition was followed by every other condition equally often[127]. These order restrictions required slightly different total trial numbers per experimental condition (48-51 trials) across the entire experiment. We minimized dependencies between the different trial phases within trials by choosing, from 1000 design matrices with randomized delays, the sequence that had the smallest correlations between the predictors in a General Linear Model (GLM) that contained the three factors of our experimental design. We used 4 different randomization protocols (sequence and timing) across the participant sample.

## Experimental setup
Participants lay supine in the MR scanner with their head stabilized by foam cushions. The right hand was cushioned into a fixed position on the belly at the left-right body midline, with the index finger extended toward the feet. Experimental instructions were displayed on a monitor, projected onto a translucent screen in the scanner bore. Participants saw the projectors' image above their head through a mirror mounted on the head coil. Accordingly, participants directed their gaze upward and could not see their hands and feet. An infrared LED was taped to the index finger to record finger movements with a video camera placed in a window between the MR and control rooms. To improve the visibility of the IR LED, cardboard was placed above the right wrist, just below the chest, to shield light emitted from the projector at the end of the scanner bore.

After three blocks of a recording session, the experimenter passively moved the participant's legs into the required uncrossed or crossed position. Participants chose before the beginning of the experiment which leg should be on top, based on comfort. The crossed

posture was fixed using cushions. If necessary for comfort, the top leg was slightly elevated using additional cushions placed below the knee. Direct skin contact of legs and feet in the crossed posture was prevented using clothing and towels to avoid the generation of conductor loops.

Experimental protocols were synchronized with volume acquisition (TR = 1880 ms) and controlled via the software Presentation, version 17.0 (Neurobehavioral Systems, Albany, USA). The fMRI experiment was conducted in two sessions, each about 2 h long including 1 h of preparation and 1 h of scanning. The two scanning sessions were scheduled between several hours and 3 weeks apart. All experimental conditions were spread equally across the two scanning sessions. Participants completed a total of 12 experimental runs of 8 min duration each. Participants rested between runs and continued with the next run whenever they were ready.

## Task practice
Participants practiced the task in a regular lab a few days before the first scanning session. Participants lay supine in a reclinable chair, and the visual display was mounted above the head, to mimic the fMRI environment. During practice, we monitored eye movements electrically with an EOG setup. Participants practiced the task until they executed the pointing movement at the correct time during the trial, aimed at the correct location, and kept the hand still at all other times. Furthermore, participants fixated a centrally presented cross throughout the task and practiced keeping the body and the head still during pointing movements.

## Tactile stimulation
For tactile foot stimulation, we applied 2 ms electrical pulses through custom-built electrodes attached to a constant current electrical stimulator (DS7A; Digitimer, Hertfordshire, United Kingdom). A manual switch (DSM367; Digitimer, Hertfordshire, United Kingdom) ensured that only a single location could be electrically stimulated at any time. One LED was placed near each of the four switch positions. At any time, one LED was illuminated and indicated the currently required electrode, and an experimenter continually adjusted the switch accordingly.

Before the experiment, the experimenter adjusted the current intensity separately for each electrode. A first electrode was adjusted, starting at 30 mV and increasing in steps of 30 mV until the participant reliably detected stimulation. Afterwards, the experimenter adjusted the threshold across the four stimulus locations until the participant indicated that all four stimuli felt equally strong and could all clearly be detected. We asked participants between runs whether tactile stimuli were still clearly detectable and increased stimulus intensity if it felt weak, or decreased it if stimulation became uncomfortable over time. On average, the selected stimulation intensity was 268.33 mV (range: 60–600 mV).

## Eye tracking
Participants had to maintain fixation throughout the experiment to avoid confounding effects of saccade planning and execution. We monitored the right eye's fixation with an fMRI-compatible eye tracker (Eye Link; SR Research, Ottawa, Canada) operated at a frame rate of 250 Hz. Eye tracker calibration used a 9-point fixation procedure before each scanning session. We identified saccades offline as a 2 s.d. deviation from the trial's mean eye position that also exceeded 20 pixels (0.5° of visual angle) on the presentation monitor. We excluded trials containing saccades from the fMRI data analysis.

## Hand movement tracking
We recorded finger movements through a window that directly faced the scanner bore from the control room, using an IR-sensitive video camera operated at a frame rate of 40 Hz. In addition to the IR LED on

the participant's finger, two color LEDs placed directly in front of the camera signaled the location of the tactile stimulus (left vs. right foot) and the instructed movement (pro vs. anti-movement) on each trial, allowing us to align video with fMRI data offline. We extracted finger position and movement direction (left vs. right, as seen from the participant's viewpoint) from the video with a custom-made, semi-automated procedure that was based on a combination of cluster detection methods, gradual averaging, and subtraction of images and automatically detected the changes of finger position across frames. Finger movements were assessed for correct left/right direction but not for the specific medial/lateral targets on each foot as it was not possible to clearly distinguish these movements due to the close distance of the medial and lateral targets.

## Data selection

We acquired 6303 trials after excluding one run of one participant due to a recording error. We excluded trials if the pointing movement was executed toward the wrong goal location (4.7%), when saccades occurred (10.2%) or when a combination of any of these occurred (0.5%). Hand movement data was unavailable or of low quality such that hand movements could not be determined in 34.0% of trials, for instance due to adverse lighting conditions or elevated leg position. Such adverse conditions often affected an entire run (49.3% of missing trials). The number of trials where hand movements could not be assessed was similar across conditions (range across conditions: 15–19 trials per condition per participant). A 2 (left vs right tactile stimulation) x 2 (uncrossed vs crossed posture) x 2 (pro vs anti task) within-subjects repeated measures ANOVA showed no main effect or interaction of any of the three factors on the number of trials where hand movements could not be reliably assessed.

Eye movement data was unavailable or low quality in 20.8% of trials, usually due to obstruction by the head coil. 7.6% of trials had unavailable or low-quality data for both hand and eye movements. Given the comparably low number of errors in trials for which all data was available, we did not exclude trials due to missing hand and eye data. The final analysis was based on 5396 trials, that is, 85.6% of all recorded trials. The number of trials included in the final analysis was similar across conditions, with an average of 42.2 trials per condition per participant (range across conditions: 41.1–43.1 trials per condition per participant).

## FMRI acquisition

We obtained fMRI data with a 3-tesla MR scanner (Siemens, Erlangen) and 32-channel head coil, using an echo planar imaging (EPI) T2*-sensitive sequence that acquired 32 axial slices in descending order (in-plane voxel size: 3 ×3 mm; slice thickness: 3 mm; slice gap: 0.51 mm; TR: 1880 ms; TE: 30 ms; flip angle: 70°; FOV: 216 × 216 mm). fMRI scans covered the whole brain with the exception of the cerebellum and a small ventral portion of the anterior temporal lobe.

## Structural MRI acquisition

We obtained a high resolution (1 × 1 x 1 mm) structural MRI image of each participant using a T1* sensitive MPRAGE sequence with 240 slices, for use in coregistration and normalization of functional data.

## Preprocessing of imaging data

We preprocessed and analyzed fMRI data with the software Statistical Parametric Mapping (SPM12, Statistical Parametric Mapping; http://www.fil.ion.ucl.ac.uk/spm/), integrated into MATLAB R2015a (The MathWorks, Natick, USA). We discarded the first four volumes of each run to allow for spin saturation. We corrected fMRI data for susceptibility artifacts and rigid body motion by unwarping and alignment to the first image of the first session. Then, we corrected functional images for differences in acquisition time, and co-registered the

individual T1 image to the mean functional image generated during realignment.

## Multi-voxel pattern classification analysis

We used a multi-voxel pattern analysis (MVPA) approach to determine whether multivariate patterns of brain activation allow classifying different types of spatial codes used during tactile processing and movement planning. This approach involved determining which brain regions differentiate pairs of characteristics, such as left vs. right foot stimulation or left vs. right-side movement goal. Figure 2, Fig. S1 and Fig. S2 illustrate the pairwise comparisons we conducted to decode tactile location (6 classifiers), movement goal location (1 classifier) and task rule (1 classifier).

We constructed participant-level GLMs that were optimized for subsequent classification. Spatial interpolations inherent in normalization and smoothing might degrade meaningful activation patterns[128]. Therefore, we calculated GLMs in non-normalized, participant space and without smoothing. We accounted for baseline drifts within runs by applying a high-pass filter (128 s), and for serial dependency within runs by using an autoregressive model. GLMs included 23 predictors that modeled experimentally induced variance of the measured BOLD signal in each voxel with delta functions marking the onsets of the particular delay, convolved, in turn, with the canonical hemodynamic response function. There were 2 baseline predictors, one for each foot posture (uncrossed, crossed) that modeled fixation delays at the beginning of each trial, as well as during the rest periods at the beginning and end of each run. Four predictors modeled the touch localization delay (uncrossed, crossed foot posture x stimulation of left, right foot). Eight predictors modeled the planning and movement execution delays (uncrossed, crossed foot posture x stimulation of left, right foot x pro-, anti-movement). Finally, all trial phases that contained behavioral errors were assigned to a common predictor. Recall, that foot posture was either uncrossed or crossed in a given run. Therefore, any given run contained only 12 (1 baseline, 2 tactile locations, 4 planning, 4 execution, 1 error) of the 23 experiment-wide predictors. We did not include head motion parameters as nuisance regressors, as our unwarping and realignment preprocessing included a correction of the fMRI images for movement-related artefacts[129], but see Fig. S4 for results with motion parameters included.

We performed MVPA with The Decoding Toolbox[128] version 3.997, using run-wise β-images for the different trial phases and experimental conditions estimated by the GLM that reflected the voxel-wise amplitude of the hemodynamic response function. We used an L2-norm support vector machine (SVM) as classifier in the implementation of LIBSVM[130], with a fixed cost of c = 1. For whole-brain unbiased voxel selection, we applied a spherical searchlight with a radius of 4 voxels[60]. On each classification fold, the classifier was trained with input patterns from run-wise β-images to differentiate between two classes, such as movement goal left vs. movement goal right. Note that the two decoded classes were always present on both uncrossed and crossed runs, thus balancing any overall response differences between different postures. Classifier performance was validated with a leave-one-out cross validation design that predicted each of the 12 runs after training based on the respective 11 other runs. We report the mean decoding accuracy across all classification iterations, depicted at the center voxel of a given searchlight region, as the measure of the overall generalization performance of the classifier of that searchlight region. We repeated this procedure for all recorded voxels, resulting in a whole-brain map of averaged decoding accuracy for the two tested conditions across all test runs. For group-level statistical analysis, we normalized participant-level accuracy maps to MNI space based on the transformation parameters obtained during segmentation of the T1 image, and applied 6 mm Gaussian kernel smoothing. Across participants and for each classifier, we used the SnPM13 toolbox to test which

brain voxels contained accuracy values that differed significantly from chance-level (50%) using a one-sample $t$-test[131,132]. Group-level tests covered the whole-brain except for the cerebellum and a small ventral portion of the anterior temporal lobe. Significance was determined using whole-brain cluster-based permutation tests using an initial threshold of $p < 0.001$ and a secondary family-wise error (FWE) correction rate of $p < 0.05$[133]. Voxels that survive this correction for multiple comparisons indicate locations where the decoding accuracy of the classifier is significantly better than chance for the differentiation between the two tested conditions. Put differently, multivariate patterns surrounding the identified voxels contain information about the differentiation between the two classes of interest, which is interpreted as their feature representation[60,66,134–137]. Visualizations of significantly higher than chance classification accuracy are based on mapping of activations identified in the 3D brain volume onto an inflated atlas of the cortical surface (Conte-69 atlas)[138] using the Computerized Anatomical Reconstruction and Editing Toolkit (Caret)[139].

### Regions of interest (ROI)

We visualized decoding accuracy by adapting an approach that allows for inter-individual differences regarding anatomical and functional brain organization[61,140]. First, we defined regions of interest (ROI) based on the group-level t-map of significant above-chance decoding accuracy. For distinct local clusters, ROIs were created as a sphere with 6 mm radius centered on the voxel with the highest statistical significance for decoding. For large clusters, we created several ROIs based on peaks within the cluster, theoretical considerations, and delineation of regions suggested by previous studies. Second, we extracted, for each individual participant, the decoding accuracy of all voxels contained in each respective group ROI and identified the voxel with the highest decoding accuracy. We created participant-specific ROIs around individual peak voxels, again sphere-shaped with a 6 mm radius. Some ROIs overlapped partially, and were therefore combined whenever they spanned a similar anatomical location and showed similar means and variances of decoding accuracy across subjects. Group mean, individual means and bootstrapped confidence intervals of decoding accuracy were calculated with the Hmisc package[141] in R[142] and visualized using ggplot2[143]. We determined the structural brain regions that were associated with each ROI using the anatomy toolbox for SPM[144] and labeled them based on previous studies related to goal-directed reaching and pointing[109].

### Reporting summary

Further information on research design is available in the Nature Portfolio Reporting Summary linked to this article.

## Data availability

The processed data generated in this study have been deposited in an OSF repository available at [https://doi.org/10.17605/OSF.IO/5BN2V]. The raw data are protected and are not available due to subject confidentiality requirements. The Conte-69 atlas is available as part of the Caret software package available at: https://sites.wustl.edu/vanessenlab/resources/. The ROI classification accuracy data generated in this study are provided in the Source Data file. Source data are provided with this paper.

## Code availability

Code is available at: https://doi.org/10.17605/OSF.IO/5BN2V.

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

## Acknowledgements

We thank Selina Pradel and Lara Pleil for help with data acquisition. This work was funded by an Emmy Noether grant of the German Research Foundation (DFG) to T.H. (He 6368/1-1, 1-2, 1-3) and a German Research Foundation (DFG) grant to T.H. (He 6368/4-1) in the DFG/ANR program for German-French Projects in the Natural, Life, and Engineering Sciences. WPM was supported by the Netherlands Organization for Scientific Research for this work (NWO-VICI: 453-11-001) and is currently supported by NWA-ORC- 1292.19.298.

## Author contributions

J.K. contributed to study design, data acquisition, planning and execution of the analysis, figure creation, and writing of the manuscript. She was responsible for data acquisition. C.F. contributed to data analysis, figure creation, and writing of the manuscript. W.P.M. contributed to study design, planning of the analysis, and writing of the manuscript. T.H. contributed to study design, planning of the analysis and writing of the manuscript, and supervised the study.

## Competing interests

The authors declare no competing interests.
