## [Peer Review File · Nature Communications]

Reviewers' Comments:

Reviewer #2:

Remarks to the Author:

Klautke and colleagues investigated, using fMRI-based MVPA, how goal locations for movements (pointing) towards touch on one's own body are represented, both during touch localization, and during movement execution. They report that touch locations are represented both in an anatomical and an external reference frame in the parietal cortex, and that these regions dynamically shift towards representations of movement goal locations once the movement target has been identified. This work enhances our current understanding of the underlying processes by combining touch localization and movement planning, and distinguishing between these processes, in the same study, and should therefore be of high significance for the field.

Overall, the study uses an elegant experimental design to dissociate between anatomical and external locations, as well as location and planning phases. Data analyses are appropriate, comprehensive and exhaustive, and the results interpreted carefully.

I have one concern/question regarding the selection of trials. In the experimental setup, the authors relied on video recordings of IR LED attached to the finger of the participants. The authors report that in 34% of all trials, hand movements could not be determined (line 661-662).

However, given the high accuracy in the remaining trials, it was assumed that these trials were also performed correctly (line 665-667), and therefore included in the analysis. Is it possible that movement errors by the participants could have contributed to the difficulties in determining these hand movements? While I have experienced some difficulties in recording movements and can imagine additional problems in cramped spaces such as an MRI machine, the report would likely benefit from additional analyses to support the authors' assumption. Specifically, how are these trials distributed over the various conditions, and would the authors obtain similar results if these trials were excluded (besides loss in power)?

Furthermore, I have a small number of minor questions and concerns:

- (1) How was the brain coverage of the fMRI data at the group level in the analysed sample? Especially with respect to lower brain regions, such as temporal and occipitotemporal cortex?
- (2) If I understand correctly, participant-level GLMs did not include any nuisance regressors, such as motion parameters? If not, why?
- (3) How was pointing performance/accuracy measured, and did the authors differentiate between "medial" and "lateral" targets on each foot? Was there feedback for the participants (during experiment or practice) regarding the precise location, given that this minor manipulation was used to "encourage participants to plan precise pointing movements" (line 561)?
- (4) What was the threshold for "reliable" detection of the (touch) stimulation?

Finally, I think the discussion might benefit from a more elaborated discussion of the task specificity of the observed effects. While the authors acknowledge that some effects (e.g. representation of touch in external space) may be induced by the task (line 444), there were additional aspects that might be task related:

- (1) In everyday situations, we often find ourselves not only moving the effector (e.g. our hand) towards a body part identified by touch (e.g. an itchy spot on the arm), but a combined movement of effector and target towards each other. Would the authors expect that, in such a situation, they would observe the same dynamics and drop of tactile-sensory coding once the movement target (i.e. part of the skin that should be scratched) is identified?
- (2) To what extent do the results depend on the choice of using pointing to indicate the touch, rather than reaching towards/touching the location?

Line 382: Here, hand-centered reference frame came to my mind as an alternative option, as in the current study, the target was always located either left or right of the participant's hand, in one line with head, gaze, and trunk

Reviewer #3:

Remarks to the Author:

One of the main goals of the work described in 399088_0_art_file_7084523_rlc3y3 (entitled "Dynamic spatial coding in parietal cortex mediates hand movement planning towards touch" by Klautke and Colleagues) is understanding the role of the posterior parietal cortex (PPC) in integrating sensory and motor signals when humans plan hand movements towards tactile targets.

Unless immobilized in rather natural postures, one of the challenges people may face during performance of such tasks is that locations of targets on our bodies and their locations in external reference frames can dissociate. In the latter case, potential targets will cross the body midline, the usual "left vs. right [body-part location]" rule will be violated, and there will inevitably be a need for it to be overwritten for effective motor control. This would be the case, however, only if external location was always a critical factor, and there was an obligatory remapping of tactile targets into external space. (See comments below and consider an open question if this is really the case if movements are directed towards body parts, regardless of their transient locations.)

By means of multi-voxel pattern analysis (MVPA), the reviewed work with the use of functional magnetic resonance imaging (fMRI) shows that the mere location of tactile targets is coded anatomically in rostral subdivisions of PPC, whereas their spatial location was coded in more caudal subdivisions of PPC. Moreover, but not surprisingly, after specification of movement goal locus – including anti-pointing target locations – the coding was purely spatial and invoked the fronto-parietal networks, bilaterally. The Authors conclude that "PPC flexibly updates its spatial codes to accommodate rule-based transformation of arbitrary sensory input to generate movement to environment and own body alike". (BTW, the 'own body alike' is doubtful here, as well; see comments below).

In general, this study seems to be well designed, capitalizing on the earlier paradigms wherein there is an alteration of the spatial position of the limbs, i.e., when body-related responses and/or targets are held either in an uncrossed or crossed configurations with respect to body midline. There is also an additional, well-established, though in somewhat different paradigms or for different effectors, arbitrary rule-based transformation included in the task, i.e., an anti response. Importantly, fMRI contrasts / analyses are set for different trial phases, and the related tasks, to reveal differences between anatomical and/or spatial codes, as well as the ones accommodating rule-based transformations for external targets. Yet, because the univariate approach to fMRI data analysis did not allow to distinguish between anatomical and spatial coding, the Authors used a more sophisticated – the aforementioned MVPA – approach. The latter did not seem to be introduced post hoc as the acquisition of the data looks optimized for its use, too. As such, it is a valuable study.

There are a few "spoilers", as well as weak points, that may reduce the appreciation of this work by both experts, and general readers. Only the glaring ones will be listed below.

Apparently, the logic and order of arguments in the Abstract is flawed, and the resulting confusion strengthened by the phrase from the title "hand movement planning towards touch". While the third sentence of the abstract "... pointed towards or away from memorized tactile stimuli on their feet, dissociating sensory and motor locations" refers to one of the most critical manipulations, it is hardly related to the required integration and interplay described in the preceding two sentences, and title. It would make more sense after a description of "crossed or uncrossed [...] feet to dissociate stimulus location relative to anatomy versus in external space". Yet, even then it is not the sensory input that is arbitrary (see "rule-based transformation of arbitrary sensory input") but the movement planning rule itself. Neither is the hand movement planned towards touch.

Even though the introduction is quite informative, with many paragraphs really well written, it contains similar conceptual flaws, listing "three dissociable spatial codes in reaches to tactually stimulated locations on one's body: (1) somatotopic and (2) external-spatial locations of the tactile stimulus, and (3) a motor goal location". Point three is irrelevant when "reaches [are performed] to tactually stimulated locations". As the last sentence of the Introduction clearly states, this study is about "movement coding to [...] tactually-defined [...] targets", but not necessarily "sensory targets"! While these subtle conceptual mistakes may seem trivial, they have consequences: the rule-based transformation could also lead to "encod[ing of] proprioceptive [...] targets in a body-centred code". No doubts, the Authors think otherwise, and in the study conclusion write "Once the movement task rule was specified, tactile information was no longer detectable, and instead a frontoparietal network encoded the location of the movement goal."

To summarize these two paragraphs, the Abstract and Introduction must be really finetuned, and conceptual mistakes removed. Otherwise, the reasoning is inconsistent, and some statements

(including selected statements from the Discussion) contradict each other (or exclusively the ones from Introduction). The Discussion itself is quite informative.

As indicated earlier, the acquisition of the data seemed to be optimized for MVPA analyses, with disparate task phases synchronized with the beginning of TRs. Task intervals varied between 1-4 TRs. As the Authors admit, they "did not identify any regions that showed differences in univariate activation to tactile stimulation on the left vs right foot, to tactile stimulation on the left vs right side of external space or between pro- and anti-pointing conditions." This is the case most likely because processing of stimulus locations or planning movements towards these locations are unlikely to extend beyond the 2nd TR, i.e., to last through the 3rd or 4th TR, i.e., 3760-5640-7520 ms. Did the Authors search for the optimal basis sets for the use in their hemodynamic response function(HRF convolution) modeling in their linear modelling of univariate activation? At any rate, no specific information on this analysis could be found in the manuscript.

I suspect that The Decoding Toolbox for MVPA takes the real voxel-wise amplitude of the hemodynamic response function, and not the theoretical one that would, for example, be predicted for all four TRs? If this is not the case, would you gain anything by reducing the number of modeled TRs to just one in all cases, or one for the shortest and just two in the case of the longer ones?

While the number of trials per participant included in the analyses seems quite sufficient, it is stated in the Methods that "The analyzed sample consisted of 16 students" (and there are indeed 16 points in the plots). No prior sample size, nor power estimations for this study are provided, however. This is a bit worrying because there are also the following pieces of information: "Three further participants were excluded from analysis: one had performed >99% of all movements to the incorrect movement goal in the crossed, but not in the uncrossed foot posture, and MR slice positioning accidentally omitted part of SPL for the other two". Why would you go for 19 participants (not 20 or 26?) at first?

Manuscript No.: NCOMMS-22-46617

Title: Dynamic spatial coding in parietal cortex mediates tactile-motor transformation

We thank the reviewers for their helpful and constructive comments. We have revised our manuscript based on their comments. Below we give details on how we have revised the manuscript to address each of the comments individually. The reviewers' original comments are shown in blue, our responses are in black and we specify the page and line numbers of revised sections in the manuscript.

Responses to Reviewer 1:

Klautke and colleagues investigated, using fMRI-based MVPA, how goal locations for movements (pointing) towards touch on one's own body are represented, both during touch localization, and during movement execution. They report that touch locations are represented both in an anatomical and an external reference frame in the parietal cortex, and that these regions dynamically shift towards representations of movement goal locations once the movement target has been identified. This work enhances our current understanding of the underlying processes by combining touch localization and movement planning, and distinguishing between these processes, in the same study, and should therefore be of high significance for the field.

Overall, the study uses an elegant experimental design to dissociate between anatomical and external locations, as well as location and planning phases. Data analyses are appropriate, comprehensive and exhaustive, and the results interpreted carefully.

We thank the reviewer for their positive appreciation of our manuscript.

R01-01:

I have one concern/question regarding the selection of trials. In the experimental setup, the authors relied on video recordings of IR LED attached to the finger of the participants. The authors report that in 34% of all trials, hand movements could not be determined (line 661-662). However, given the high accuracy in the remaining trials, it was assumed that these trials were also performed correctly (line 665-667), and therefore included in the analysis. Is it possible that movement errors by the participants could have contributed to the difficulties in determining these hand movements? While I have experienced some difficulties in recording movements and can imagine additional problems in cramped spaces such as an MRI machine, the report would likely benefit from additional analyses to support the authors' assumption. Specifically, how are these trials distributed over the various conditions, and would the authors obtain similar results if these trials were excluded (besides loss in power)?

Thank you for pointing out this concern. The difficulty that hand movements could not always be determined was typically due to technical difficulties such as adverse lighting conditions or the participant's posture interfering with the detection of the infrared LED on their finger. To give an example, participants' knees were slightly bent, so it could happen that the hand was positioned low and the knees obstructed the view of the LED on the video image. Accordingly, if this was the case, often an entire run of trials was affected, which was corrected before the next run was started. We found no link between condition type and the number of missing trials. We have added details describing this in the methods section (**page 27-28, lines 754-762**).

Page 27-28, lines 754-762: *Hand movement data was unavailable or of low quality such that hand movements could not be determined in 34.0% of trials, for instance due to adverse lighting conditions or elevated leg position. Such adverse conditions often affected an entire run (49.3% of missing trials). The number of trials where hand movements could not be assessed was similar across conditions (range across conditions: 15 – 19 trials per condition per participant). A 2 (left vs right tactile stimulation) x 2 (uncrossed vs crossed posture) x 2 (pro vs anti task) within-subjects repeated measures ANOVA showed no main effect or interaction of any of the three factors on the number of trials where hand movements could not reliably be assessed.*

To further ensure that trials lacking hand movement information did not affect our decoding results, we re-ran our analyses on a dataset that excluded all trials where hand movements could not be determined. Note that this analysis was thus based on less training data for the MVPA classifiers, because some runs had to be completely excluded (see above, entire runs being affected by finger visibility problems). Fig. S5 presents the results of the new analyses, showing clusters both corrected for multiple comparisons ($p < 0.05$ FWE corrected) and uncorrected for multiple comparisons ($p < 0.01$). At the uncorrected threshold, searchlight clusters are located in the same regions as for the MVPA analyses based on all trials. Thus, results are similar regardless of whether trials without hand movement information are included or not, but power is reduced, as expected, due to the reduction in the total number of trials available for MVPA classification analyses (34% of the total number of trials removed).

R01-02:

Furthermore, I have a small number of minor questions and concerns:

(1) How was the brain coverage of the fMRI data at the group level in the analysed sample? Especially with respect to lower brain regions, such as temporal and occipitotemporal cortex?

Subject-level fMRI data and group analyses covered the whole brain with the exception of the cerebellum and a small ventral portion of the anterior temporal lobe. We chose this coverage because our main area of interest was the parietal cortex. We have specified this in the methods section on **page 28, lines 774-776** and **page 29, lines 837-839**.

R01-03:

(2) If I understand correctly, participant-level GLMs did not include any nuisance regressors, such as motion parameters? If not, why?

We did not include motion parameters as nuisance regressors in the GLM as we applied a “Realignment and Unwarp” preprocessing step in SPM, which aims to model and correct for movement-related artefacts in the fMRI time-series signal (Andersson et al., 2001). Thus, fMRI images are already corrected for movement-related variance prior to participant-level GLM modelling. We have specified this in the methods section (**page 29, lines 814-817**).

To confirm that our MVPA results were not affected by any residual movement-related signal changes, we re-ran separate instances of our MVPA analyses that included motion parameters as nuisance regressors in participant GLMs. Results are shown in Fig. S4. In all analyses, locations of clusters are consistent with the analyses without motion parameters included as nuisance regressors.

R01-04:

(3) How was pointing performance/accuracy measured, and did the authors differentiate between “medial” and “lateral” targets on each foot? Was there feedback for the participants (during experiment or practice) regarding the precise location, given that this minor manipulation was used to “encourage participants to plan precise pointing movements” (line 561)?

Pointing accuracy was assessed by determining the direction of the infra-red LED attached to the participant’s index finger (left or right) using a custom-made, semi-automated procedure that was based on a combination of cluster detection methods, gradual averaging, and subtraction of images and automatically detected the changes of finger position across frames. It was not possible to distinguish between pointing movements to the medial and lateral targets from the LED video recording data due to the close distance of the targets to one another. This has been specified in the text on **page 27, lines 747-749**.

Participants were not given feedback on their pointing accuracy as pointing movements were analysed offline. We have specified this in the text on **page 25, lines 657-658**.

We have used a similar manipulation in previous research, in which participants had to point towards visual stimuli with their index finger, where the visual stimuli were at different eccentricities (Heed et al. 2011, 2016, see detailed refs below). In those studies, there was always a correlation between target location and finger pointing eccentricity. We instructed participants in the same way in the present study as in these previous studies, emphasizing that they had to point to the target as precisely as possible. This was practiced prior to the fMRI recordings. Given that this procedure was

successful in our previous research, we have no reason to assume it would have failed here. We have revised the sentence that explains our rationale for using 2 locations and added pointers to the discussed previous work (**page 6, lines 127-131**).

Refs:

Heed, T., Beurze, S. M., Toni, I., Röder, B., & Medendorp, W. P. (2011). Functional Rather than Effector-Specific Organization of Human Posterior Parietal Cortex. *J. Neurosci.*, *31*(8), 3066–3076. <https://doi.org/10.1523/JNEUROSCI.4370-10.2011>

Heed, T., Leone, F. T. M., Toni, I., & Medendorp, W. P. (2016). Functional versus effector-specific organization of the human posterior parietal cortex: Revisited. *Journal of Neurophysiology*, *116*(4), 1885–1899. <https://doi.org/10.1152/jn.00312.2014>

R01-05:

(4) What was the threshold for “reliable” detection of the (touch) stimulation?

Stimulation intensity was adjusted individually for each participant. For each electrode, current intensity began at 30 mV and was increased in steps of 30 mV until the participant reliably detected stimulation – that is, when a stimulus was administered by hand in the control room, the participant responded on each occasion that the stimulus had been detected. The experimenter then adjusted the threshold across the four stimulus locations until the participant indicated that all four stimuli felt equally strong and could all clearly be detected – that is, the participant indicated for all stimuli during the final intensity adjustment that they were clearly perceivable, did not miss a stimulation, and at the same time confirmed that stimulation was painless. Thus, by reliable we mean that no stimulus was missed. We did not attempt to adjust stimuli to have a specified intensity above a psychophysically detected detection threshold; this seemed irrelevant because stimuli were clearly suprathreshold. Accordingly, there was no incidental report of any participant that they had the impression that there were trials without stimulation. On average, the selected stimulation intensity was 268.33 mV (range: 60-600 mV). See **page 27, lines 721-728**.

R01-06:

Finally, I think the discussion might benefit from a more elaborated discussion of the task specificity of the observed effects. While the authors acknowledge that some effects (e.g. representation of touch in external space) may be induced by the task (line 444), there were additional aspects that might be task related:

(1) In everyday situations, we often find ourselves not only moving the effector (e.g. our hand)

towards a body part identified by touch (e.g. an itchy spot on the arm), but a combined movement of effector and target towards each other. Would the authors expect that, in such a situation, they would observe the same dynamics and drop of tactile-sensory coding once the movement target (i.e. part of the skin that should be scratched) is identified?

Thank you for raising this point. Clearly, in everyday life, participants would be very likely to move not just the reaching effector but also the target area. Being able to move both the acting and target effectors is, indeed, a special case of somatosensory targets. As a comparison, in the more “traditional” visuo-motor paradigms the target is a visual object that the participant cannot control. Here, we aimed to keep the design as close to visuomotor paradigms as possible to aid comparison between our results and these studies. We are not aware of studies that have investigated the coding of a touched limb and effector moving to each other, but we consider it an important question for future research. We have noted this point, and its potential effect on maintained sensory coding in the discussion (**page 22, lines 551-560**).

R01-07:

(2) To what extent do the results depend on the choice of using pointing to indicate the touch, rather than reaching towards/touching the location?

We have added a paragraph in the discussion (**page 22, lines 542-551**) to discuss potential differences between pointing and reaching to touch the body.

Page 22, lines 542-551: In natural behavior, movements in response to tactile stimuli often involve directly touching the stimulated location or limb. For example, one might brush the right hand across the left arm after feeling an insect crawling along it. Our study differs from such natural situations and behavior in some regards. First, in our study, participants pointed towards, rather than moved to, the locations indicated by touch. fMRI work has compared responses for pointing and reaching and observed similar activation for both in a frontoparietal network consistent with the regions in which we decoded the pointing movement goal in the present study¹¹⁹. Notably, touching the own skin gives rise to additional somatosensory feedback that has been shown to improve tactile localization¹²⁰, and there can be attenuation of somatosensory signals due to predictive mechanisms^{121,122}.

R01-08:

Line 382: Here, hand-centered reference frame came to my mind as an alternative option, as in the current study, the target was always located either left or right of the participant’s hand, in one line with head, gaze, and trunk

We agree that a hand-centred reference frame can also be considered, as was already mentioned in the Introduction (**page 3, lines 38-43**), and is now also mentioned in the discussion (**page 19, lines 407-410**). The distinction of hand-, eye/gaze-, and trunk-centered concerns only the origin of an otherwise identical type of 3D(-like) spatial coding system. In contrast, the skin is 2D. Therefore, transformation from skin to any of the 3D reference frames requires additional processing to consider how the skin is wrapped around the body – in contrast to transformations between the different 3D reference frames. To avoid too much complexity (and, along with it, an even longer experiment duration), we didn't include any manipulations to dissociate the different possible anchors for the 3D reference frame.

Responses to Reviewer 2:

One of the main goals of the work described in 399088_0_art_file_7084523_rlc3y3 (entitled “Dynamic spatial coding in parietal cortex mediates hand movement planning towards touch” by Klautke and Colleagues) is understanding the role of the posterior parietal cortex (PPC) in integrating sensory and motor signals when humans plan hand movements towards tactile targets. Unless immobilized in rather natural postures, one of the challenges people may face during performance of such tasks is that locations of targets on our bodies and their locations in external reference frames can dissociate. In the latter case, potential targets will cross the body midline, the usual “left vs. right [body-part location]” rule will be violated, and there will inevitably be a need for it to be overwritten for effective motor control. This would be the case, however, only if external location was always a critical factor, and there was an obligatory remapping of tactile targets into external space. (See comments below and consider an open question if this is really the case if movements are directed towards body parts, regardless of their transient locations.)

By means of multi-voxel pattern analysis (MVPA), the reviewed work with the use of functional magnetic resonance imaging (fMRI) shows that the mere location of tactile targets is coded anatomically in rostral subdivisions of PPC, whereas their spatial location was coded in more caudal subdivisions of PPC. Moreover, but not surprisingly, after specification of movement goal locus – including anti-pointing target locations – the coding was purely spatial and invoked the fronto-parietal networks, bilaterally. The Authors conclude that “PPC flexibly updates its spatial codes to accommodate rule-based transformation of arbitrary sensory input to generate movement to environment and own body alike”. (BTW, the ‘own body alike’ is doubtful here, as well; see comments below).

In general, this study seems to be well designed, capitalizing on the earlier paradigms wherein there is an alteration of the spatial position of the limbs, i.e., when body-related responses and/or targets are held either in an uncrossed or crossed configurations with respect to body midline. There is also an additional, well-established, though in somewhat different paradigms or for different effectors, arbitrary rule-based transformation included in the task, i.e., an anti response. Importantly, fMRI contrasts / analyses are set for different trial phases, and the related tasks, to reveal differences between anatomical and/or spatial codes, as well as the ones accommodating rule-based transformations for external targets. Yet, because the univariate approach to fMRI data analysis did not allow to distinguish between anatomical and spatial coding, the Authors used a more sophisticated – the aforementioned MVPA – approach. The latter did not seem to be introduced post hoc as the acquisition of the data looks optimized for its use, too. As such, it is a valuable study.

There are a few “spoilers”, as well as weak points, that may reduce the appreciation of this work by both experts, and general readers. Only the glaring ones will be listed below.

We thank the reviewer for their overall positive feedback on our manuscript and the pointers to some very important and interesting theoretical aspects of the processes we investigated.

In this passage, the reviewer hints at a further comment on our conclusion that PPC mediates movements “to the own body alike” (marked yellow by us), but then doesn’t directly address the point anymore later on. We’ll briefly reply to it here. We point out that our task was always related to a specific location on the own body. This was the case for both pro and anti-pointing. In the pro case, instructions were to point precisely to the location of the tactile stimulus. In the anti-case, the instruction was to point to the same location on the other foot.

We have reformulated the abstract to make the task more obvious and emphasized this point in Results (**page 6, line 133-135**).

R02-01:

Apparently, the logic and order of arguments in the Abstract is flawed, and the resulting confusion strengthened by the phrase from the title “hand movement planning towards touch”. While the third sentence of the abstract “... pointed towards or away from memorized tactile stimuli on their feet, dissociating sensory and motor locations” refers to one of the most critical manipulations, it is hardly related to the required integration and interplay described in the preceding two sentences, and title. It would make more sense after a description of “crossed or uncrossed [...] feet to dissociate stimulus location relative to anatomy versus in external space”. Yet, even then it is not the sensory input that is arbitrary (see “rule-based transformation of arbitrary sensory input”) but the movement planning rule itself. Neither is the hand movement planned towards touch.

Even though the introduction is quite informative, with many paragraphs really well written, it contains similar conceptual flaws, listing “three dissociable spatial codes in reaches to tactually stimulated locations on one’s body: (1) somatotopic and (2) external-spatial locations of the tactile stimulus, and (3) a motor goal location”. Point three is irrelevant when “reaches [are performed] to tactually stimulated locations”. As the last sentence of the Introduction clearly states, this study is about “movement coding to [...] tactually-defined [...] targets”, but not necessarily “sensory targets”! While these subtle conceptual mistakes may seem trivial, they have consequences: the rule-based transformation could also lead to “encod[ing of] proprioceptive [...] targets in a body-centred code”. No doubts, the Authors think otherwise, and in the study conclusion write “Once the movement task rule was specified, tactile information was no longer detectable, and instead a frontoparietal network encoded the location of the movement goal.”

To summarize these two paragraphs, the Abstract and Introduction must be really finetuned, and conceptual mistakes removed. Otherwise, the reasoning is inconsistent, and some statements (including selected statements from the Discussion) contradict each other (or exclusively the ones from Introduction). The Discussion itself is quite informative.

Thank you for raising these conceptual issues with our reference to movement planning towards touch. We have altered the text where such references were made, in particular the title, abstract, introduction and discussion, to better clarify these points. The title is now “Dynamic spatial coding in parietal cortex mediates tactile-motor transformation”. In the abstract we have changed the order

of sentences as suggested by the reviewer and correctly specified the pointing task. In the introduction we have clarified that the reaches are made to tactually-defined locations based on the task rule.

Incidentally, we actually do not necessarily “think otherwise” – we’re rather undecided. We had written the paper from a traditional viewpoint in which touch location is thought to be recoded into some kind of 3D spatial location. However, our recent research has highlighted a number of cases in which this is apparently not the case (e.g. Badde et al. 2019, Maij et al. 2020). However, these studies had not involved a traditional “aim to a sensory target” type of paradigm. Yet, in a new in-press paper we show that adaptation effects for tactile features, such as motion direction, are not external-spatial but depend on the body’s default posture (Badde & Heed, 2023). In the discussion of that paper, we make exactly the suggestion made here by the reviewer, namely that touch may never be recoded into 3D space but, instead, is referred to a limb and any movements then aim for that limb.

We think that this idea poses new problems in that we are well able to plan movements (pointing and reaching alike) to arbitrary locations on a limb. This means that a matching of 2D skin to 3D space must occur at least on the level of a single limb, if it does not happen in an overarching touch-to-space “representation” or transformation.

Our study doesn’t solve this distinction – and wasn’t designed to. Nevertheless, we now discuss these points in the revised manuscript’s Discussion as one potential mechanism of tactile-motor coding.

Refs:

Badde, S., Röder, B., & Heed, T. (2019). Feeling a Touch to the Hand on the Foot. *Current Biology*, 29(9), 1491-1497.e4. <https://doi.org/10.1016/j.cub.2019.02.060>

Maij, F., Seegelke, C., Medendorp, W. P., & Heed, T. (2020). External location of touch is constructed post-hoc based on limb choice. *ELife*, 9, e57804. <https://doi.org/10.7554/eLife.57804>

Badde, S., & Heed, T. (2023). The hands’ default location guides tactile spatial selectivity. *PNAS*, 120(15), e2209680120. <https://doi.org/10.1073/pnas.2209680120>

R02-02:

As indicated earlier, the acquisition of the data seemed to be optimized for MVPA analyses, with disparate task phases synchronized with the beginning of TRs. Task intervals varied between 1-4 TRs. As the Authors admit, they “did not identify any regions that showed differences in univariate activation to tactile stimulation on the left vs right foot, to tactile stimulation on the left vs right side of external space or between pro- and anti-pointing conditions.” This is the case most likely because processing of stimulus locations or planning movements towards these locations are unlikely to extend beyond the 2nd TR, i.e., to last through the 3rd or 4th TR, i.e., 3760-5640-7520 ms. Did the Authors search for the optimal basis sets for the use in their hemodynamic response function(HRF convolution) modeling in their linear modelling of univariate activation? At any rate, no specific information on this analysis could be found in the manuscript.

I suspect that The Decoding Toolbox for MVPA takes the real voxel-wise amplitude of the hemodynamic response function, and not the theoretical one that would, for example, be predicted for all four TRs? If this is not the case, would you gain anything by reducing the number of modeled TRs to just one in all cases, or one for the shortest and just two in the case of the longer ones?

Thank you for raising this suggestion. We now ran the respective analyses by modelling all trial phases with only 1 TR, independent of the true, jittered 1-4 TR duration with which a given trial part had been presented. We present detailed results of these analyses in the supplemental material and Figures S6 and S7. While results were mostly in line with what we had reported based on the full-TR analyses, the new analysis decoded external-spatial location bilaterally, rather than unilaterally. This finding is important because it may resolve the inconsistency of our study with most other reported results on external coding of touch in PPC. We discuss this new result in the Discussion now (**page 20, lines 467-473**). Moreover, we point towards the detailed analysis in the Results (**page 16, lines 337-342**).

Note, that we have decided to keep the original analysis in the main paper. We modelled the total duration of the sensory processing or movement planning interval for each of our conditions in our participant-level GLMs and then input the resulting beta-images of the resulting fit of these models to the fMRI data as input for MVPA analyses using The Decoding Toolbox (see Hebart et al., 2015 for details of the benefits of this MVPA approach). This approach models a sustained sensory or movement coding that is kept in memory. Our main rationale for presenting this model is that it is common for studies using delayed movement paradigms to model the full interval duration (for example see Beurze et al., 2007, 2009, 2010; Gertz & Fiehler, 2015; Heed et al., 2011). Moreover, neuronal recordings support the idea of sustained neuronal firing for movement planning during a delay period (e.g. Gail & Andersen, 2006). Therefore, our results using this approach aid comparison of our results to those of previous studies.

R02-03:

While the number of trials per participant included in the analyses seems quite sufficient, it is stated in the Methods that “The analyzed sample consisted of 16 students” (and there are indeed 16 points in the plots). No prior sample size, nor power estimations for this study are provided, however. This is a bit worrying because there are also the following pieces of information: “Three further participants were excluded from analysis: one had performed >99% of all movements to the incorrect movement goal in the crossed, but not in the uncrossed foot posture, and MR slice positioning accidentally omitted part of SPL for the other two”. Why would you go for 19 participants (not 20 or 26?) at first?

We booked scanning slots for 20 participants, and during the scanning of the first pilot participant, a technical error caused only partial data to be collected during the first session for this participant. As the participant did not complete the two sessions, we did not mention them in the methods section. To avoid this confusion, we have modified the description of the participants in the Methods section to mention this additional participant (**page 25, lines 633-637**). Due to time and funding constraints,

it was not feasible to arrange further scanning slots and acquire further participants to replace the participants whose data could not be analysed.

Reviewers' Comments:

Reviewer #1:

Remarks to the Author:

My concerns have been sufficiently addressed and resolved by the authors.

Reviewer #2:

Remarks to the Author:

In the revised manuscript, now entitled "Dynamic spatial coding in parietal cortex mediates tactile-motor transformation", the Authors have satisfactorily addressed all my concerns. This was the case despite the fact that there was some uncertainty about whether or not one of the main critical points of my review was developed sufficiently enough later on to be adequately tackled. I have no further comments.

Gregory Króliczak

Adam Mickiewicz University in Poznań